

# Factors controlling dissolved $^{137}$Cs activities in Matsukawa-ura lagoon, a semi-closed estuary, after the Fukushima accident

Takuya Niida[1,2], Hyoe Takata[3], Sho Watanabe[4], Shinya Namura[2], Toshihiro Wada[3]

[1] Graduate School of Symbiotic Systems Science and Technology, Fukushima University, 1 Kanayagawa, Fukushima City, Fukushima 960-1296, Japan
[2] Laboratory for Instrumentation and Analysis, Environmental Engineering Division, KANSO TECHNOS CO., LTD, 3-1-1, Higashikuraji, Katano City, Osaka 576-0061, Japan
[3] Institute of Environmental Radioactivity, Fukushima University, 1 Kanayagawa, Fukushima City, Fukushima 960-1296, Japan
[4] Fukushima Prefectural Research Institute of Fisheries Resources, 1-1-14 Koyo, Soma City, Fukushima 970-0005, Japan

*Correspondence to*: Takuya Niida (niida_takuya@kanso.co.jp)

**Abstract.** The spatial and seasonal dynamics of $^{137}$Cs were investigated from 2021 to 2023 in Matsukawa-ura lagoon, a semi-closed estuarine area approximately 40 km north of the Fukushima Daiichi Nuclear Power Plant, Japan. Weighted mean dissolved $^{137}$Cs concentrations in the lagoon waters ranged from 5.3 to 19 Bq m$^{-3}$, 2.4–8.6 times higher than those in the surrounding coastal seawater and inflowing river waters. Mass balance calculation suggests that the dissolution of $^{137}$Cs from bottom sediments sustained the high dissolved $^{137}$Cs concentrations in the estuarine water. Furthermore, dissolved $^{137}$Cs concentrations in the lagoon were higher in summer (July) than in winter (February). Quantification of the source revealed that dissolution caused $^{137}$Cs concentrations in bottom sediments to decrease by more than 2.4% over 30 days in July, but only by <0.8% over 30 days in February. This finding indicates that warmer waters during the summer accelerate the dissolution of $^{137}$Cs from bottom sediments.

Dependence of dissolved $^{137}$Cs concentration on water temperature in Matsukawa-ura lagoon and the source of $^{137}$Cs.

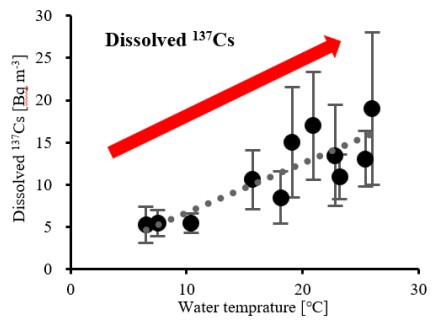

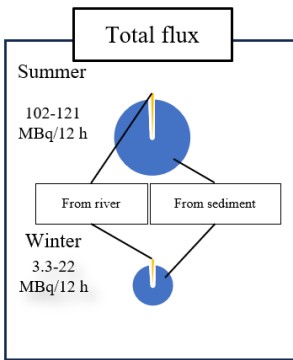





## 1 Introduction

The Fukushima Dai-ichi Nuclear Power Plant (FDNPP) accident on 11 March 2011 released important amounts of radioactive Cs ($^{134}$Cs and $^{137}$Cs) into the surrounding areas and the North Pacific. It is estimated that a total of 15–20 PBq of

$^{137}$Cs was released into the atmosphere between 12 March and 30 April 2011, with 10%–40% (2–6 PBq) estimated to have been deposited in eastern Japan (Aoyama et al., 2016) [1]. Currently, the dissolved $^{137}$Cs activity concentration in seawater more than 30 km offshore Fukushima has returned to pre-accident levels (Kusakabe and Takata, 2020) [2], whereas that in coastal waters of Fukushima Prefecture remains above pre-accident levels (Suzuki et al, 2022) [3]. Potential sources of dissolved $^{137}$Cs in Fukushima coastal waters include direct inflow from the FDNPP, leaching from seabed sediments, and

inflow from rivers. The completion of impermeable seaside wall in 2016 may have recently limited direct inflow from the FDNPP (Machida et al., 2019) [4]. However, seawater infiltrating bottom sediments leaches out adsorbed $^{137}$Cs, adding it to the seawater (Sanial et al., 2017) [5]. Indeed, Otosaka et al. (2020) [6] estimated the concentration of dissolved $^{137}$Cs in pore waters within seabed sediments to be 10–40 times higher than that in the overlying water (seawater approximately 60 cm above the seafloor), suggesting that the leaching of radioactive Cs from sediments to pore water is a $^{137}$Cs source in coastal

areas.

Extensive studies of the riverine transport of $^{137}$Cs from land to estuaries have revealed that most of the $^{137}$Cs transported from land to ocean is in the particulate phase (e.g., Nagao et al., 2013 [7]; Yamashiki et al., 2014 [8]; Niida et al., 2022 [9]). Although the proportion of dissolved $^{137}$Cs supplied by rivers is extremely small, dissolved $^{137}$Cs concentrations in the marine environment tend to be higher at near-shore sites (e.g., river mouths) than in offshore waters (Takata et al., 2020a) [10].

Accordingly, the supply from rivers to the marine environment is considered to increase dissolved $^{137}$Cs concentrations. This supply is mainly regulated by water temperature and the competition between particulate-bound $^{137}$Cs and ions in seawater, as described below.

Recent studies in rivers have suggested that the distribution coefficient between particulate-bound $^{137}$Cs and dissolved $^{137}$Cs decreases with increasing water temperature, making it easier for $^{137}$Cs to be released from suspended particles in rivers

during the warmer summer season (Igarashi et al., 2022 [11]; Tsuji et al., 2023 [12]). Furthermore, Machida et al. (2019) [4] estimated the $^{137}$Cs outflow from the harbor of the FDNPP and reported higher levels in summer than in winter, indicating that the dissolved $^{137}$Cs concentrations in the harbor may be related to water temperature.

Experiments reproducing the interaction between dissolved and particulate $^{137}$Cs due to the flow of particulate-bound $^{137}$Cs from rivers into the sea have shown that the distribution coefficient ($K_d$) between particulate-bound $^{137}$Cs and dissolved $^{137}$Cs

decreases along a salinity gradient (Li et al., 1984 [13]; Turner, 1996 [14]). These results suggest that $^{137}$Cs$^+$ can be desorbed from the particles due to competition with ions such as K$^+$ and NH$_4^+$ (Takata et al., 2020b [15], 2021 [16]).

The relationships between water temperature and dissolved $^{137}$Cs concentration in river water and between salinity and dissolved $^{137}$Cs in seawater are often discussed (Takata et al., 2022 [17]; Tsuji et al., 2023 [12]), but those in estuarine areas have





not been sufficiently addressed. One of the reasons for this is that the dissolved $^{137}$Cs transported from land and leached from
sediments is immediately diluted and dispersed into large amounts of seawater, making quantitative assessments challenging.

This study focuses on Matsukawa-ura lagoon (Soma City, Fukushima Prefecture) and its inflowing rivers to discuss the
supply of $^{137}$Cs to the lagoon and the spatial and seasonal dynamics of $^{137}$Cs within the lagoon. Matsukawa-ura lagoon is a
semi-closed estuarine area approximately 40 km north of the FDNPP, providing an ideal area for estimating the flux of $^{137}$Cs
transported from rivers and desorbed from sediments. Additionally, the lagoon is only connected to the Pacific Ocean
through a 100-m-wide mouth at its northmost point, facilitating the quantification of the mass balance of $^{137}$Cs within the
lagoon. The aim of this study was to investigate the distribution of $^{137}$Cs inputs, in turn allowing us to evaluate the
contribution of $^{137}$Cs supplied by rivers to estuarine areas, the relationships with salinity and water temperature, and the
contribution of dissolution of $^{137}$Cs from bottom sediments. Our results improve our understanding of radioactive
contamination in aquatic habitats.


## 2 Material and Methods

### 2.1 Study Area and Sampling Stations

Matsukawa-ura lagoon is a semi-closed estuarine area with an area of 6.48 km$^2$ with fluctuating salinity and water
temperature conditions (Wada et al., 2011 [18]; Noda et al., 2021 [19]). The mean water temperature in the lagoon during 1991–
2021 was 15.1 °C, with a maximum of 27.1 °C and a minimum of 5.6 °C (Fukushima Prefecture [20]). Four rivers flow into
Matsukawa-ura lagoon: Koizumi River (catchment area 17.8 km$^2$), Uda River (100.6 km$^2$), Ume River (10.7 km$^2$), and
Nikkeshi River (22.6 km$^2$). According to the fourth aerial survey conducted by the Ministry of Education, Culture, Sports,
Science and Technology (November 2011) [21], the mean inventory of deposited $^{137}$Csin the Koizumi, Uda, Ume, and
Nikkeshi catchments were 70, 205, 70, and 93 kBq m$^{-2}$ respectively, with relatively higher concentrations observed in the
forested areas of the upstream Uda catchment. The mean concentration of $^{137}$Cs across the entire Matsukawa-ura catchment
was 163 kBq m$^{-2}$ (Figure 1a). Due to the impermeable bedrock in the midstream to upstream areas of the lagoon's watershed,
it is considered that precipitation hardly infiltrates the underground, instead directly flowing into the rivers and delivering
52.1% of the total precipitation runoff to the lagoon through the rivers (Kamo et al., 2014 [22]). Additionally, Arita et al.
(2014) [23] estimated the total accumulation of $^{137}$Cs in surface sediments (0–20 cm depth) within the lagoon to be 220 GBq as
of November 2013.

In this study, we conducted 11 samplings from June 2021 to February 2023 at 13 sites including downstream sites in the
four rivers, sites at the mouths of three rivers (Koizumi, Uda, and Ume Rivers), five sites within the lagoon (from the
southeast to near the lagoon mouth in the north), and a site 800 m offshore along the outer coast of the lagoon (Figure 1b).
Detailed sampling locations and dates are provided in Tables S1 and S2. Table 1 presents estimated flow rates for each river
on each sampling date, calculated based on the Thiessen polygon method determined by the locations of Japan





Meteorological Agency observation stations and assuming that 52.1% of precipitation flows into the rivers (Kamo et al., 2014 [22]).

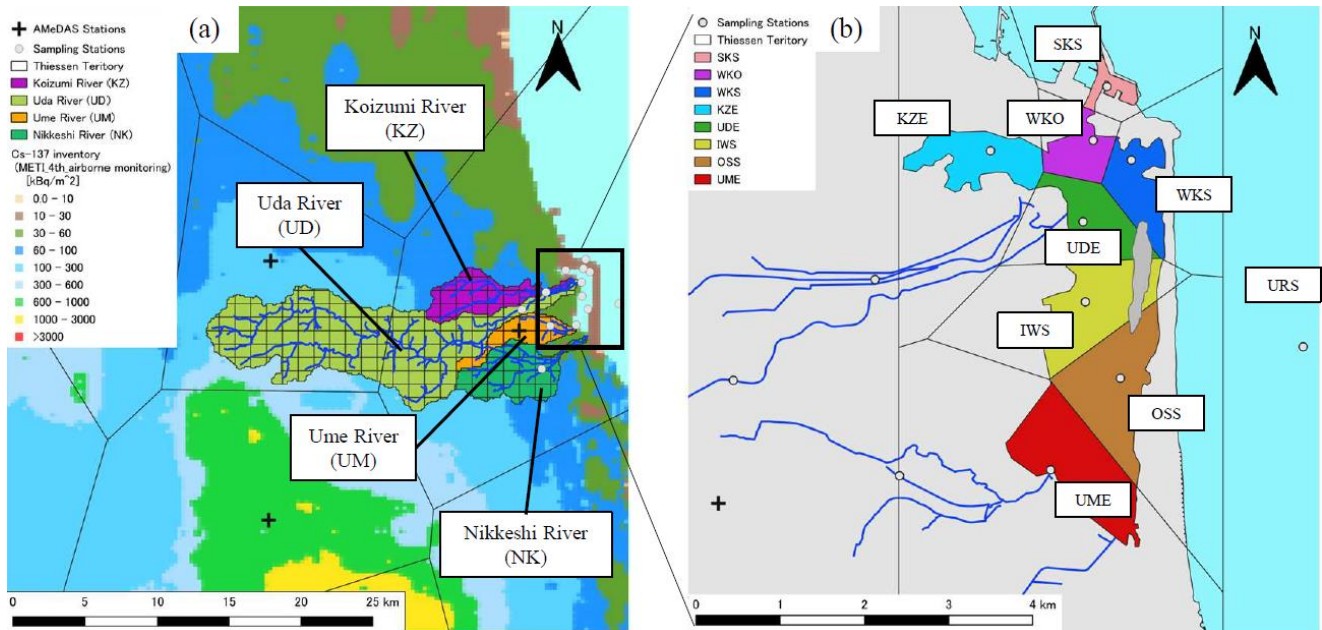

**Figure 1: Spatial distribution of the [137]Cs inventory in the study catchment (a) and sampling stations of Matsukawa-**
**ura lagoon (b). The spatial distribution of the [137]Cs inventory is based on the fourth airborne survey by MEXT (2011).**
**The Thiessen territories were created based on the coordination of Japan Meteorological Agency weather stations (a)**
**and sampling stations (b).**

## 2.2 Sample Processing and Analysis

At each sampling site, 30–40 L of river water or surface seawater were collected using 10 L polyethylene buckets. The collected water samples were transferred to 20 L polyethylene containers and brought to the laboratory. A portion of each sample was used to measure water temperature and electrical conductivity, from which salinity was calculated. On each sampling day, the water temperature in the lagoon was measured by using a chlorophyll turbidity sensor (ACLW2-CAD, JFE Advantech Co., Ltd, Hyogo, Japan) at the mouth of the lagoon at 10:00 JST. Water samples were filtered using a 0.45 µm membrane filter (047-MFPES045, AS ONE Corporation, Osaka, Japan), and approximately 20 L of the filtrate were stored for dissolved [137]Cs analysis. Additionally, 1–2 L of each sample were filtered through pre-weighed 0.4 µm polycarbonate filters (16040004, ADVANTEC, Tokyo, Japan) to measure the suspended particle concentration. The filters used for filtering 30–40 L of water were air-dried for about one week at 30 °C, then placed in 100 mL polyethylene containers for measurement of [137]Cs radioactivity in the suspended particles (Bq kg$^{-1}$-dry) using a non-destructive gamma-ray spectrometer with a coaxial high-purity Ge detector (HPGe) (GEM40, SEIKO EG&G, Tokyo, Japan). The results were then divided by the suspended particle concentration (mg L$^{-1}$) to calculate the radioactivity of [137]Cs in particulate form (Bq



m$^{-3}$). The detection limits for $^{137}$Cs ranged from 4.5 to 1590 Bq kg$^{-1}$-dry for measurement times of 80,000 s to 300,000 s, respectively. The counting efficiencies of these HPGe semiconductor detectors were calibrated using volume standard sources (MX033U8PP, The Japan Radioisotope Association, Tokyo, Japan).

To analyze dissolved $^{137}$Cs, we followed the method reported Aoyama et al. (2013) [24], summarized here. The filtrate
stored for dissolved $^{137}$Cs analysis was adjusted to a pH of approximately 1.6 with 15 M HNO$_3$. Then, 0.39 g of CsCl was added as a carrier and stirred for 2 h. Subsequently, Cs was coprecipitated with 6 g of ammonium phosphomolybdate (AMP, KANSO TECHNOS Co., LTD, Osaka, Japan). The activity of radiocesium present as an impurity in the AMP was 0.05 mBq/g-AMP. The Cs-AMP precipitate was left overnight to settle, then filtered using a paper filter with a pore size of 1 µm. After air-drying the filter for about one week at 30 °C, the precipitate was enclosed in a 10 mL Teflon container and its
weight yield was determined gravimetrically. Yields exceeded 90% for all samples. The Cs-AMP compounds enclosed in the Teflon containers were measured using a non-destructive gamma-ray spectrometer with a well-type high-purity Ge detector (GWL-90-15, SEIKO EG&G, Tokyo, Japan), and the result was reported as dissolved $^{137}$Cs radioactivity (Bq m$^{-3}$). The detection limit for dissolved $^{137}$Cs was less than 2 Bq m$^{-3}$ for all samples. The radioactivity of $^{137}$Cs was decay-corrected to the sampling date.

## 3 Results and Discussion

### 3.1 $^{137}$Cs Concentrations in River Waters

Suspended particle concentrations (mg L$^{-1}$), particulate $^{137}$Cs concentrations (Bq m$^{-3}$), $^{137}$Cs concentrations in suspended particles (Bq kg$^{-1}$-dry), dissolved $^{137}$Cs concentrations (Bq m$^{-3}$), and apparent distribution coefficients ($K_d$, L kg$^{-1}$) in the rivers flowing into the lagoon from 2021 to 2023 are shown in Figure 2. All riverine $^{137}$Cs measurement results are listed in
Table S1.

The mean suspended particle concentrations were 6.7, 7.3, 14.4, and 7.5 g m$^{-3}$ in Koizumi, Uda, Ume, and Nikkeshi Rivers, respectively, with respective median values of 2.5, 1.0, 15, and 5.4 g m$^{-3}$ (Figure 2a). The mean values are markedly higher than the median values because samples taken during increased rainfall in August 2022 increased the mean values. Suspended particle concentrations in the Ume River were relatively high compared to those in the other three rivers; this was
likely due to the reduced forest floor coverage in the Ume catchment (Table 1), which is known to engender increased soil erosion (Nishikiori et al., 2015 [25]). Similarly, the median particulate $^{137}$Cs concentrations in the Koizumi, Uda, Ume, and Nikkeshi Rivers were 3.0, 2.0, 16, and 3.7 Bq m$^{-3}$, respectively (Figure 2b). Increased suspended particle concentrations are related to increased particulate $^{137}$Cs concentrations (Ueda et al., 2013 [26]), and both tend to increase during high flow conditions (Nagao et al., 2013 [7]; Niida et al., 2022 [9]). Therefore, the higher particulate $^{137}$Cs concentrations in the Ume River
were due to increased suspended particle concentrations in the river water.

The mean $^{137}$Cs concentrations in suspended particles in the Koizumi, Uda, Ume, and Nikkeshi Rivers were 1656, 1871, 2048, and 1224 Bq kg$^{-1}$, respectively (Figure 2c). Although the mean inventory of deposited $^{137}$Cs in the Uda catchment was



2–3 times higher than those in the other catchments, the mean $^{137}$Cs concentration in suspended particles in the Uda River was less than twice those in the other rivers.

The mean dissolved $^{137}$Cs concentrations in the Koizumi, Uda, Ume, and Nikkeshi Rivers were 2.2, 1.7, 2.6, and 3.5 Bq m$^{-3}$, respectively (Figure 2d). The Uda River showed the lowest dissolved $^{137}$Cs concentration among the four rivers, similar to the distribution reported previously (Takata et al., 2022 [17]).

In catchments with high soil $^{137}$Cs concentrations, the $^{137}$Cs concentration in suspended particles and the dissolved $^{137}$Cs concentration flowing into the rivers tended to be high. We note, though, that the actual relationship is influenced by many 145    factors such as topography, vegetation, rainfall patterns, and soil properties; accordingly, the relation between the mean soil $^{137}$Cs concentration and $^{137}$Cs in riverine particles or dissolved is not necessarily simply proportional. In downstream areas of catchments such as the Uda River, where soil $^{137}$Cs concentrations are high in upstream areas and low in downstream areas, high $^{137}$Cs concentrations in suspended particles and dissolved $^{137}$Cs concentrations transported from the upstream area through the river may be diluted by downstream river waters containing less $^{137}$Cs (Yamashiki et al., 2014 [8]).

$K_d$ values ranged from $7.2 \times 10^4$ to $3.7 \times 10^6$ L kg$^{-1}$ (Figure 2e), similar to those of rivers elsewhere in Fukushima Prefecture between 2011 and 2014, which ranged from $7.7 \times 10^4$ to $1.4 \times 10^6$ L kg$^{-1}$ (Taniguchi et al., 2019 [27]).

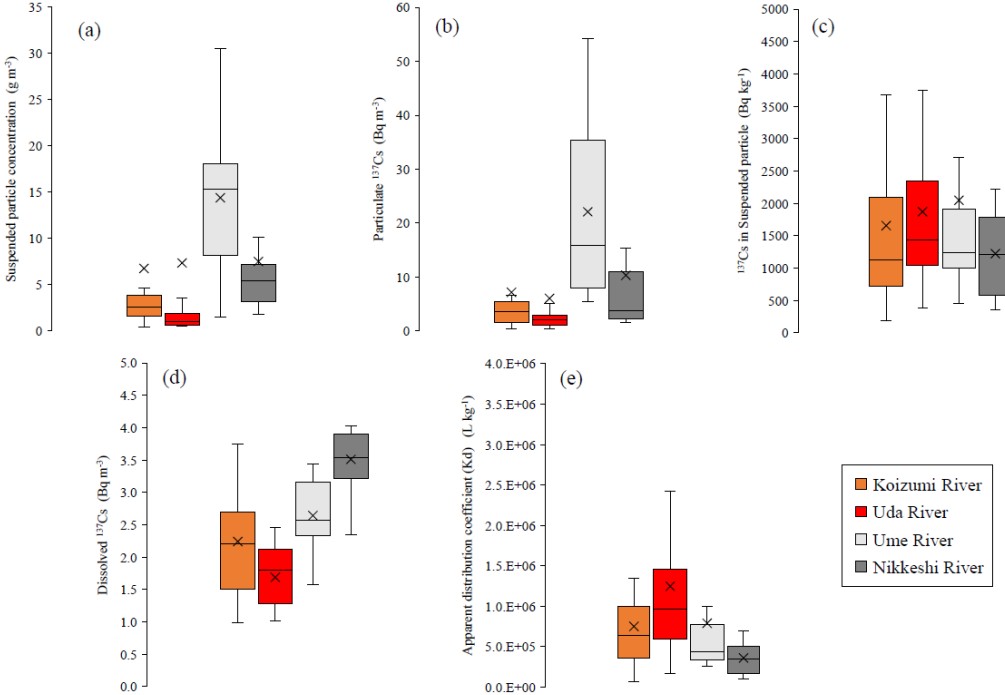

**Figure 2: Water conditions in the four in flowing rivers: (a) suspended particle concentration, (b) particulate $^{137}$Cs concentration, (c) $^{137}$Cs concentration in suspended particles, (d) dissolved $^{137}$Cs concentration, and (e) the**
**distribution coefficient (Kd). Box plots represent the median and interquartile values, and the whiskers show the minimum and maximum values. Cross marks represent the arithmetic means of the results.**



**Table 1: Catchment means rainfall and flux of river water discharge.**

| | | Weather station | | Catchment | | | | |
| --- | --- | --- | --- | --- | --- | --- | --- | --- |
| | | Hippo | Soma | Koizumi | Uda | Ume | Nikkeshi | Total |
| Catchment area (km²) | | | | 17.8 | 100.6 | 10.7 | 22.6 | 151.6 |
| Land use (%) | Forest | | | 51 | 85 | 27 | 62 | |
| | Paddy field | | | 18 | 5 | 41 | 22 | |
| | Farmland | | | 6 | 6 | 6 | 4 | |
| | Urban area | | | 22 | 1 | 20 | 3 | |
| Ratio to entire catchment (%) | Koizumi river (KZ) | 0 | 100 | | | | | |
| | Uda river (UD) | 60 | 40 | | | | | |
| | Ume river (UM) | 0 | 100 | | | | | |
| | Nikkeshi river (NK) | 0 | 100 | | | | | |
| Rainfall in the 30 days prior to sample collection (mm) | Jun-21 | 91 | 74 | 74 | 84 | 74 | 74 | |
| | Jul-21 | 165 | 197 | 197 | 177 | 197 | 197 | |
| | Aug-21 | 346 | 321 | 321 | 336 | 321 | 321 | |
| | Sep-21 | 211 | 195 | 195 | 204 | 195 | 195 | |
| | Oct-21 | 181 | 177 | 177 | 179 | 177 | 177 | |
| | Dec-21 | 117 | 64 | 64 | 96 | 64 | 64 | |
| | Feb-22 | 21 | 12 | 12 | 17 | 12 | 12 | |
| | Jun-22 | 217 | 235 | 235 | 224 | 235 | 235 | |
| | Aug-22 | 141 | 99 | 99 | 124 | 99 | 99 | |
| | Oct-22 | 54 | 97 | 97 | 71 | 97 | 97 | |
| | Feb-23 | 47 | 28 | 28 | 39 | 28 | 28 | |
| Flux of river water discharge (×10⁶ m³/12 h)[a] | Jun-21 | | | 0.011 | 0.073 | 0.0068 | 0.014 | 0.11 |
| | Jul-21 | | | 0.030 | 0.15 | 0.018 | 0.039 | 0.24 |
| | Aug-21 | | | 0.050 | 0.29 | 0.030 | 0.063 | 0.44 |
| | Sep-21 | | | 0.030 | 0.18 | 0.018 | 0.038 | 0.26 |
| | Oct-21 | | | 0.027 | 0.16 | 0.016 | 0.035 | 0.23 |
| | Dec-21 | | | 0.010 | 0.083 | 0.0059 | 0.013 | 0.11 |
| | Feb-22 | | | 0.0019 | 0.015 | 0.0011 | 0.0024 | 0.020 |
| | Jun-22 | | | 0.036 | 0.20 | 0.022 | 0.046 | 0.30 |
| | Aug-22 | | | 0.015 | 0.11 | 0.0092 | 0.019 | 0.15 |
| | Oct-22 | | | 0.015 | 0.062 | 0.0090 | 0.019 | 0.10 |
| | Feb-23 | | | 0.0043 | 0.034 | 0.0026 | 0.0055 | 0.047 |

[a] Based on the data from Kamo et al.(2014).

## 3.2 $^{137}$Cs Concentrations in Matukawa-ura Lagoon

### 3.2.1 Relationship Between Dissolved $^{137}$Cs and Salinity

The relationships between suspended particle concentrations, particulate $^{137}$Cs concentrations, $^{137}$Cs concentrations in suspended particles, dissolved $^{137}$Cs concentrations, $K_d$, and salinity in rivers, within Matsukawa-ura lagoon, and in the coastal area are shown in Figure 3. The mean salinity in the lagoon during the study period was 29.1. All salinity and $^{137}$Cs measurements in the lagoon and coastal seawater (station URS) are listed in Table S2.

The mean and median suspended sediment concentrations in the lagoon were 10.8 and 3.2 g m$^{-3}$, respectively (range 0.3–208 g m$^{-3}$). The mean and median particulate $^{137}$Cs concentrations were 8.1 and 2.3 Bq m$^{-3}$, respectively (range 0.1–123 Bq m$^{-3}$). The mean and median $^{137}$Cs concentrations in suspended particles were 1111 and 631 Bq kg$^{-1}$, respectively (range 100–





16,434 Bq kg$^{-1}$). Previous study has reported that these concentrations tend to decrease with increasing salinity (Takata et al., 2022 [17]), which could be due to the dilution, coagulation, and settling of suspended particles along the salinity gradient as well as dilution by seawater with low $^{137}$Cs concentrations, but our study did not find clear relationships (Figure 3a, b, c).

The mean and median dissolved $^{137}$Cs concentrations in the lagoon were 9.9 and 8.2 Bqm$^{-3}$, respectively (range 2.5–31.3 Bq m$^{-3}$). In contrast to particulate $^{137}$Cs and $^{137}$Cs in suspended particles, dissolved $^{137}$Cs concentrations were higher in the lagoon than in the four inflowing rivers. Dissolved $^{137}$Cs concentrations also tended to increase with increasing salinity (Figure 3d), being the highest at salinities of 25–30, then decreasing at salinities above 30. This trend implies that the dissolved $^{137}$Cs concentration in the lagoon temporarily increases due to desorption of particulate $^{137}$Cs from the rivers (Takata et al., 2020a [10]) and the supply of dissolved $^{137}$Cs from pore water in seabed sediments (Kambayashi et al., 2021 [28]; Takata et al., 2022 [17]) but is also diluted by the large amount of coastal seawater that flows into the lagoon. However, the large variation of dissolved $^{137}$Cs concentrations at salinities of 25–30 (2.5–31.3 Bq m$^{-3}$) may be due to seasonality and differences between sampling sites, which are described in the next subsection.

The $K_d$ values tended to decrease with increasing salinity (Figure 3e). It has been observed that some amount of radiocesium adsorbed to suspended particles flowing into the ocean is desorbed from the suspended particles due to competition with ionic species such as K$^+$ and NH$_4^+$ (Takata et al., 2020a [10]; 2021 [16]), consistent with our results.

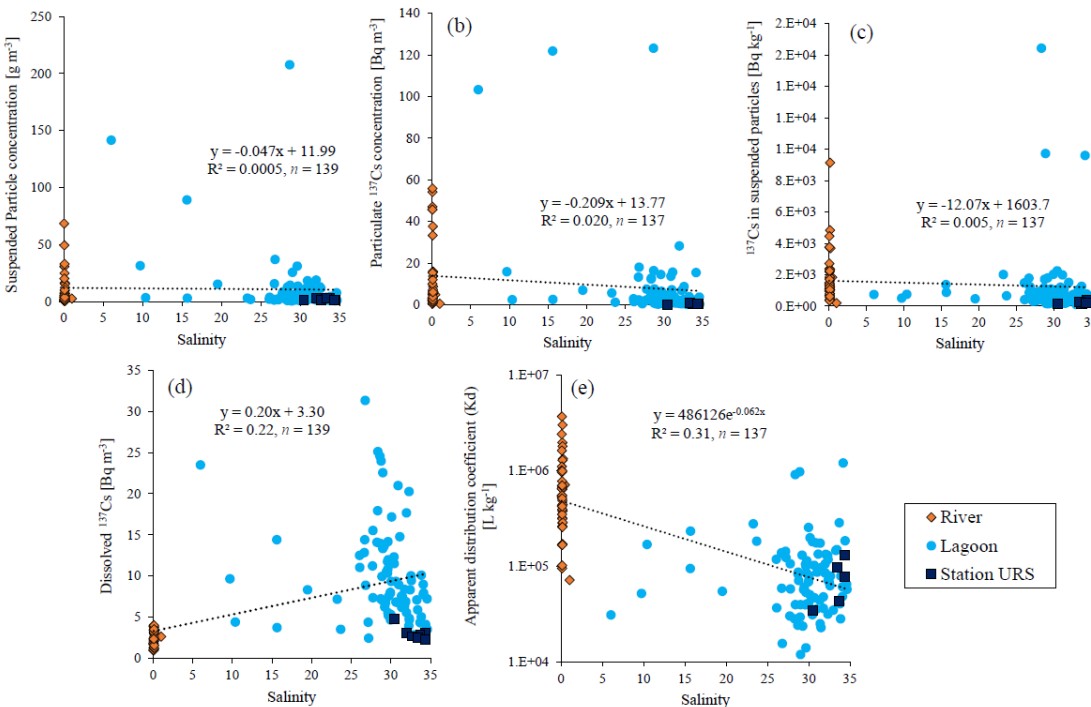

**Figure 3: (a) Suspended particle concentration, (b) particulate $^{137}$Cs concentration, (c) $^{137}$Cs concentration in suspended particles, (d) dissolved $^{137}$Cs concentration, and (e) apparent distribution coefficient (Kd) plotted against salinity in Matsukawa-ura lagoon.**



### 3.2.2 Relationship Between Dissolved [137]Cs Concentration, Salinity, and Water Temperature

The relationship between dissolved [137]Cs concentration, salinity, and water temperature at each sampling site within the lagoon is shown in Figure 4.

Dissolved [137]Cs concentrations tended to decrease to the north and in proximity to the mouth of the lagoon, being highest at sites UME, OSS, and IWS on the south side of the lagoon (Figure 4a–c, respectively). This is due to increased mixing with coastal seawater containing low [137]Cs near the lagoon mouth. Furthermore, at all sampling sites in the lagoon with salinities around 25–30, dissolved [137]Cs concentrations tended to be higher when the water was warmer (Figure 4, ≥20 °C, red and orange symbols) and lower when the water temperature was cooler (Figure 4, ≤15 °C, light blue and blue symbols).

To further analyze the relationship between dissolved [137]Cs concentration and water temperature in the lagoon, time-dependent changes in dissolved [137]Cs concentrations in the lagoon are shown in Figure 5a, and the relationship between dissolved [137]Cs concentration and water temperature is shown in Figure 5b. Lagoon-wide weighted mean dissolved [137]Cs concentrations were calculated from the sampling sites using Voronoi partitioning, taking the area of each Voronoi cell as the weight during averaging by area ratio. The weighted mean dissolved [137]Cs concentrations in the lagoon on each sampling date were 5.3–19.0 Bq m$^{-3}$, 2.4–8.6 times higher than the dissolved [137]Cs concentrations in the coastal seawater (site URS) collected during the same period (2.2–4.8 Bq m$^{-3}$).

The dissolved [137]Cs concentration in the lagoon tended to be higher in the summer and lower in the winter (Figure 5a) and showed a significant correlation with water temperature (Figure 5b). Previous studies have revealed that water temperature influences the adsorption and desorption of [137]Cs between solutions and particles with high affinities, such as clay minerals having "frayed-edge-sites" (FeS). Tertre et al. (2005) [29] evaluated the effect of temperature on Cs$^+$ behavior at low ionic strength under neutral conditions and reported that $K_d$ decreases by a factor of 3 between 25 and 150 °C. Furthermore, Igarashi et al. (2022) [11] reported a relationship between the $K_d$ value of radiocesium and water temperature in the midstream catchment of the Abukuma River, which flows through Fukushima Prefecture; they suggested that $K_d$ decreases as water temperature increases.

Here, in the studied semi-closed estuary, the dissolved [137]Cs concentration in the lagoon fluctuated greatly at salinities around 25–30. Nagao et al. (2020) [30] reported 0.1% desorption of [137]Cs from sand samples in ultrapure water; 3.7% in a 25% seawater solution; 7.1% in a 50% seawater solution; and 10%–12% in 100% seawater, in artificial seawater, and in a 470 mM NaCl + 8 mM KCl solution. In other words, it is possible that the desorption of [137]Cs is largely completed at the salinity of a 50% seawater solution. Our results suggest that both salinity and water temperature affect the adsorption and desorption of particulate [137]Cs, changing the distribution of dissolved [137]Cs in the lagoon. In Matsukawa-ura lagoon (average salinity = 29.1), the desorption of [137]Cs by ion exchange in seawater was largely complete, implying that the effect of water temperature dominates over that of salinity in the thermodynamic adsorption and desorption process in saline water.





In the next section, we use mass balance calculations to analyze the sources of dissolved ¹³⁷Cs in the lagoon and compare

220   the contributions of river inputs against the supply from seabed sediments.

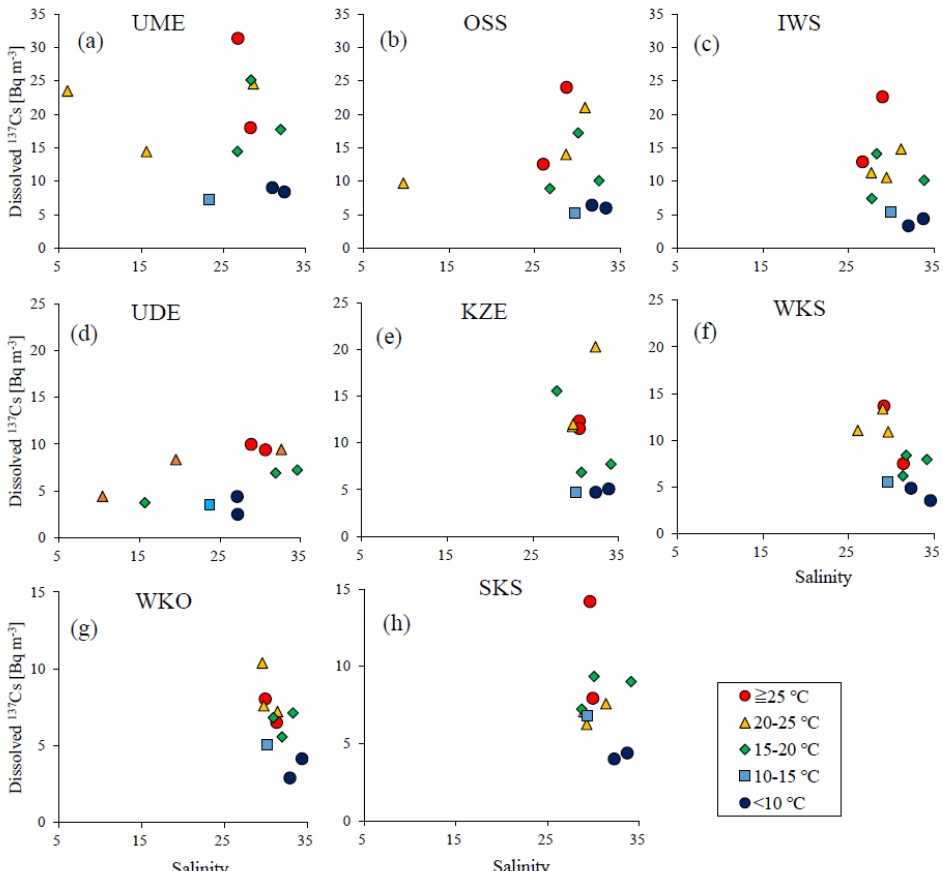

**Figure 4: Dissolved¹³⁷Cs concentration versus salinity at each sampling site in the Matsukawa-ura lagoon. Hotter and cooler colors indicate warmer and cooler water temperatures, respectively.**



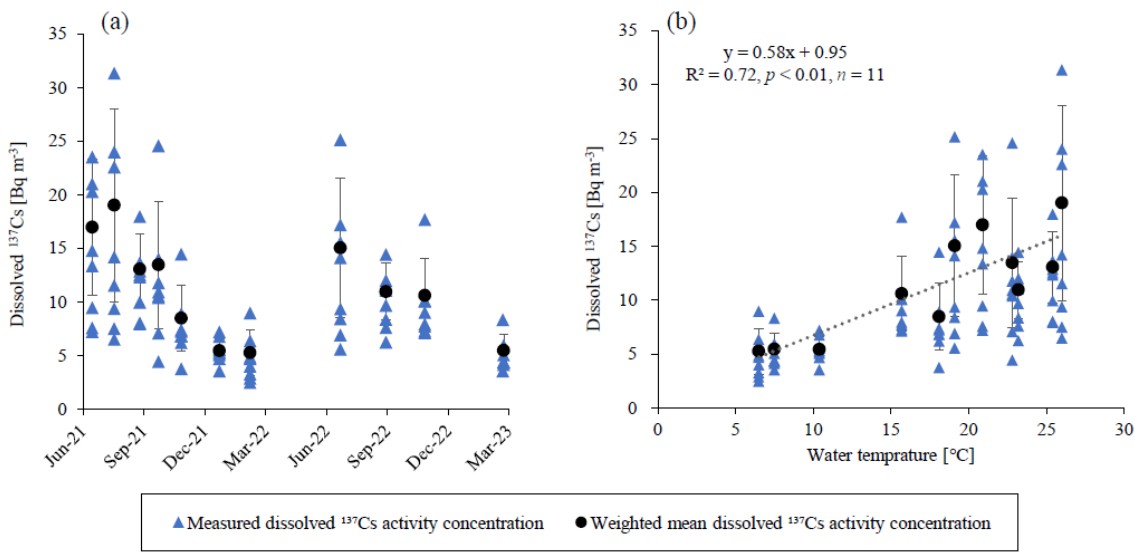

**Figure 5:** (a) Temporal variations of dissolved $^{137}$Cs concentrations and (b) dissolved $^{137}$Cs concentrations versus water temperature in Matsukawa-ura lagoon from June 2021 to February 2023. Black bars represent the standard deviation of the dissolved $^{137}$Cs concentrations on each sampling date.

### 3.3 Mass Balance Calculations

We used mass balance calculations to evaluate the magnitudes of the internal and external sources responsible for the non-conservative mixing behavior of dissolved $^{137}$Cs observed in the lagoon. The results of the calculation are shown in Table 2. Each flux was normalized to a time interval of 12 h to allow for the semidiurnal tidal periodicity in the lagoon (Kamo et al., 2014 [22]). The amount of seawater flowing into the lagoon has been estimated to be 7.2 Mm$^3$ in a 12 h period (Kamo et al., 2014 [22]).

The flux of $^{137}$Cs from each river into the lagoon was estimated by multiplying the river water discharge by the particulate and dissolved $^{137}$Cs concentrations. The $^{137}$Cs concentrations in river water on each sampling date are reported in Table S1. The flux of Cs flowing into the lagoon from the four rivers at 12-h intervals is shown in Figure S1. The combined particulate $^{137}$Cs flux from the four rivers ranged from 0.06 to 7.24 MBq/12 h (Figure S1a), the combined dissolved $^{137}$Cs flux from 0.03 to 1.23 MBq/12 h (Figure S1b), and the total (particulate + dissolved) flux from 0.17 to 7.47 MBq/12 h (Figure S1c). The mean proportion of particulate $^{137}$Cs in the total flux was about 50%–60% in all but the Ume River. However, during high flow conditions, particulate $^{137}$Cs accounted for more than 90% of the total flux in all rivers (Figure S1d), as reported in previous studies (Nagao et al., 2013 [7]; Yamashiki et al., 2014 [8]; Niida et al., 2022 [9]).





The proportion of particulate $^{137}$Cs in river water that desorb and dissolves into the water after flowing into the lagoon was assumed to be 5.5% (Takata et al., 2021 [17]), and this was calculated as desorbed $^{137}$Cs. The weighted mean dissolved $^{137}$Cs

concentration in the lagoon during the study period was 5.3–19.0 Bq m$^{-3}$, whereas the dissolved $^{137}$Cs concentration in coastal seawater outside the lagoon (station URS) during the same period was 2.2–4.8 Bq m$^{-3}$. This result indicates that mixing in river mouths in the lagoon increased the dissolved $^{137}$Cs concentration by 0.5–16.8 Bq m$^{-3}$ after coastal seawater flowed into the lagoon. Therefore, 3.4–121 MBq/12 h of dissolved $^{137}$Cs was added to the lagoon. However, the fluxes of dissolved and desorbed $^{137}$Cs from the river were 0.03–1.23 MBq/12 h and 0.005–0.077 MBq/12 h, respectively, which

cannot be explained by the supply from the rivers alone. Therefore, the supply of dissolved $^{137}$Cs from seabed sediments must be much greater than that from rivers. Kambayashi et al. (2021) [28] measured the $^{137}$Cs concentration in pore water in sediments of Matsukawa-ura lagoon in 2016 and estimated the flux from the sediments to be 139–293 MBq/day, suggesting that the supply of $^{137}$Cs from bottom sediments may account for more than 90% of the supply of $^{137}$Cs to Matsukawa-ura lagoon, which is generally consistent with our estimates.

Arita et al. (2014) [23] estimated the inventory of $^{137}$Cs concentrations in the bottom sediments of the lagoon and the total accumulation of $^{137}$Cs in the top 20 cm of sediment to be 220 GBq as of 2013. The flux of $^{137}$Cs that desorbed from sediments and dissolved into seawater in 12-h intervals, calculated based on the total $^{137}$Cs inventory of seabed sediments reported by Arita et al. (2014) [23] and the results of our survey, is 0.001%–0.07% of the total $^{137}$Cs inventory of sediments in the lagoon (Table 3).

Quantification of the sources revealed that the $^{137}$Cs concentration in bottom sediments decreased via dissolution by more than 2.4% in 30 days in July, and by only <0.8% in the same period in February. Comparing data from June 2021, December 2021, and October 2022, when precipitation amounts and $^{137}$Cs fluxes from rivers in the Matsukawa-ura catchment were similar (Tables 1 and 2), the dissolution rate from bottom sediments was high in June 2021 and low in December 2021. This finding suggests that physical factors such as continuous inflow from rivers are unlikely to strongly impact the

spatiotemporal variation of dissolved $^{137}$Cs concentrations in the lagoon. Thus, we infer that warmer water temperatures during the summer may accelerate the dissolution of $^{137}$Cs from bottom sediments.

Based on our results, we conclude that $^{137}$Cs in bottom sediments, deposited during the early stages of FDNPP accident, gradually dissolves when pore waters are exposed to seawater flowing into the lagoon. However, warmer seawater temperatures during the summer may further accelerate the dissolution process. This finding implies that the processes

observed in the lagoon have changed the spatiotemporal distribution of $^{137}$Cs in the coastal waters of Fukushima Prefecture.






**Table 2: Mass balance of dissolved $^{137}$Cs in Matsukawa-ura lagoon.**

| | Input of seawater into Matsukawa-ura lagoon[a] | Dissolved $^{137}$Cs in coastal water (station URS) | Weighted average dissolved $^{137}$Cs in lagoon | Potential flux of $^{137}$Cs | Input of river water into Matsukawa-ura lagoon | Flux of riverine dissolved $^{137}$Cs | $^{137}$Cs percentage of desortable fraction from river particles[b] | Desorable fracton of riverine particulate $^{137}$Cs | Estimated flux of $^{137}$Cs desoobed from bottom sediment |
|---|---|---|---|---|---|---|---|---|---|
| | ×10$^6$ m$^3$/12 h | Bq m$^{-3}$ | Bq m$^{-3}$ | ×10$^6$ Bq/12 h | ×10$^6$ m$^3$/12 h | ×10$^6$ Bq/12 h | % | ×10$^6$ Bq/12 h | ×10$^6$ Bq/12 h |
| Jun-21 | 7.2 | 2.2 - 4.8 | 17.0 | 88 - 106 | 0.11 | 0.27 | 5.5 | 0.036 | 87 - 106 |
| Jul-21 | 7.2 | 2.2 - 4.8 | 19.0 | 102 - 121 | 0.24 | 0.61 | 5.5 | 0.060 | 102 - 120 |
| Aug-21 | 7.2 | 2.2 - 4.8 | 13.1 | 60 - 78 | 0.44 | 1.23 | 5.5 | 0.077 | 58 - 77 |
| Sep-21 | 7.2 | 2.2 - 4.8 | 13.5 | 62 - 81 | 0.26 | 0.57 | 5.5 | 0.078 | 62 - 80 |
| Oct-21 | 7.2 | 2.2 - 4.8 | 8.5 | 27 - 45 | 0.23 | 0.38 | 5.5 | 0.086 | 26 - 45 |
| Dec-21 | 7.2 | 2.2 - 4.8 | 5.4 | 4.7 - 23 | 0.11 | 0.25 | 5.5 | 0.021 | 4.4 - 23 |
| Feb-22 | 7.2 | 2.2 - 4.8 | 5.3 | 3.4 - 22 | 0.02 | 0.03 | 5.5 | 0.0031 | 3.3 - 22 |
| Jun-22 | 7.2 | 2.2 - 4.8 | 15.0 | 74 - 93 | 0.30 | 0.64 | 5.5 | 0.053 | 73 - 92 |
| Aug-22 | 7.2 | 2.2 - 4.8 | 11.0 | 44 - 63 | 0.15 | 0.23 | 5.5 | 0.40 | 44 - 62 |
| Oct-22 | 7.2 | 2.2 - 4.8 | 10.6 | 42 - 61 | 0.10 | 0.20 | 5.5 | 0.010 | 42 - 60 |
| Feb-23 | 7.2 | 2.2 - 4.8 | 5.5 | 4.8 - 24 | 0.05 | 0.08 | 5.5 | 0.0052 | 4.8 - 23 |

[a] Kamo et al.(2014).

[b] Takata et al.(2021).

**Table 3: Estimated dissolution rates from the bottom sediment at Matsukawa-ura lagoon.**

| | Elapsed days | $^{137}$Cs amount of bottom sediment (0-20 cm depth)[a] | Estimated flux of $^{137}$Cs desorbed from bottom sediment[b] | Dissolution rates | |
|---|---|---|---|---|---|
| | day | ×10$^9$ Bq | ×10$^6$ Bq/12 h | %/12 h | %/30 days |
| Nov-13 | 0 | 220 | | | |
| Jun-21 | 2785 | 185 | 87 - 106 | 0.047 - 0.057 | 2.8 - 3.4 |
| Jul-21 | 2818 | 184 | 102 - 120 | 0.055 - 0.065 | 3.3 - 3.9 |
| Aug-21 | 2856 | 184 | 58 - 77 | 0.032 - 0.042 | 1.9 - 2.5 |
| Sep-21 | 2884 | 183 | 62 - 80 | 0.034 - 0.044 | 2.0 - 2.6 |
| Oct-21 | 2918 | 183 | 26 - 45 | 0.014 - 0.024 | 0.85 - 1.5 |
| Dec-21 | 2975 | 182 | 4.4 - 23 | 0.0024 - 0.013 | 0.14 - 0.76 |
| Feb-22 | 3021 | 182 | 3.3 - 22 | 0.0018 - 0.012 | 0.11 - 0.73 |
| Jun-22 | 3156 | 180 | 73 - 92 | 0.041 - 0.051 | 2.4 - 3.1 |
| Aug-22 | 3224 | 180 | 44 - 62 | 0.024 - 0.035 | 1.5 - 2.1 |
| Oct-22 | 3282 | 179 | 42 - 60 | 0.023 - 0.034 | 1.4 - 2.0 |
| Feb-23 | 3400 | 178 | 4.8 - 23 | 0.003 - 0.013 | 0.16 - 0.79 |

[a] Based on the data from Arita et al.(2014).

[b] Quated from Table 2.





## 4 Conclusion

We investigated the spatial and seasonal dynamics of $^{137}$Cs in Matsukawa-ura lagoon, a semi-closed estuarine area approximately 40 km north of the Fukushima Daiichi Nuclear Power Plant, Japan. The dissolved $^{137}$Cs concentrations in the lagoon were higher than that in the river water flowing into the lagoon and in the coastal seawater, suggesting the influence

of desorption from suspended sediments due to increasing salinity. Furthermore, it was found that dissolved $^{137}$Cs concentrations in the lagoon were high in summer and low in winter, suggesting influence of water temperatures.

Quantification of the sources by mass balance calculations revealed that the supply of dissolved $^{137}$Cs from bottom sediments was much greater than the supply of dissolved $^{137}$Cs from rivers and suspended sediments. These results suggest that continuous $^{137}$Cs input from rivers are unlikely to have a strong impact on the spatiotemporal variation of dissolved $^{137}$Cs

concentrations in the lagoon, and we conclude that $^{137}$Cs in the sediments deposited during the early stages of Fukushima Daiichi Nuclear Power Plant accident was exposed to seawater flowing into the lagoon and gradually dissolved. In addition, higher seawater temperatures in summer may accelerate the dissolution of $^{137}$Cs from bottom sediments. Future work should address the relationship between $^{137}$Cs in the bottom sediments and pore waters of estuaries, as well as waters in coastal areas, while taking into account seasonal variations in water temperature.

**Team list**

Corresponding Author

Takuya Niida – *Graduate School of Symbiotic Systems Science and Technology, Fukushima University, 1 Kanayagawa, Fukushima City, Fukushima 960-1296, Japan;* Email (present): s2571005@ipc.fukushima-u.ac.jp

*Laboratory for Instrumentation and Analysis, Environmental Engineering Division, KANSO TECHNOS CO., LTD, 3-1-1,*
*Higashikuraji, Katano City, Osaka 576-0061, Japan;* Phone: +81-72-810-6551; Email (permanent): niida_takuya@kanso.co.jp

Authors

Hyoe Takata - *Institute of Environmental Radioactivity, Fukushima University, 1 Kanayagawa, Fukushima City, Fukushima 960-1296, Japan;* Email: h.takata@ier.fukushima-u.ac.jp
Sho Watanabe - *Fukushima Prefectural Research Institute of Fisheries Resources, 1-1-14 Koyo, Soma City, Fukushima 970-0005, Japan;* Email: watanabe_shou_01@pref.fukushima.lg.jp
Shinya Namura - *Laboratory for Instrumentation and Analysis, Environmental Engineering Division, KANSO TECHNOS CO., LTD, 3-1-1, Higashikuraji, Katano City, Osaka 576-0061, Japan;* Email: namura_shinya@kanso.co.jp

Toshihiro Wada - *Institute of Environmental Radioactivity, Fukushima University, 1 Kanayagawa, Fukushima City,*
*Fukushima 960-1296, Japan;* Email: t-wada@ipc.fukushima-u.ac.jp

**Author contribution**

The manuscript was written through contributions of all authors. All authors have given approval to the final version of the
manuscript.

**Competing interest**

The authors declare that they have no conflict of interest.

**Acknowledgements**

We are grateful to researchers of the Fukushima Prefectural Research Institute of Fisheries Resources for their cooperation
during sampling.

**Financial support**

This research is an achievement of " Stabilizing resources through effective release of seedlings using ICT infrastructure"
(JPFR23060109, JPFR24060109, JPFR25060109) among advanced technology development projects in the field of
agriculture, forestry and fisheries. (Fukushima Institute for Research, Education and Innovation (F-REI)).

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
