# Peer review of "Factors controlling dissolved 137Cs activities in Matsukawa-ura lagoon, a semi-closed estuary, after the Fukushima accident"

_EGUsphere, 2025_

## Author Comment (AC1)

The manuscript addresses an important topic with valuable data, but significant reorganization, clearer figures, and expanded discussion are needed to improve clarity and scientific impact. I therefore recommend **major revisions**. With the above changes, the paper would be much more accessible to a broader readership and would better highlight its contributions.

We thank the reviewers for their time and constructive feedback. Please see our responses below.

**Major Comments**

**1. Figures and Visualization**

- **Figure 1:** The current layout is difficult to interpret: the legends are too small, and colors are used simultaneously to represent 137Cs deposition and river catchments. I recommend separating these elements:
  - Create a first panel where colors represent only 137Cs deposition, and identify river catchments with labels instead of colors.
  - Zoom in slightly so that Fukushima Dai-ichi Nuclear Power Plant is clearly visible.
  - Include an inset map showing the study area within Japan to orient readers unfamiliar with the region.
  - Describe all acronyms in the figure caption.
- Contextual location: There is no figure linking the lagoon to its main 137Cs source (Fukushima). A simple map indicating the relative position of the lagoon and nuclear plant would strengthen context for non-local readers.

The map has been adjusted to make it easier to see the relative positions of the study area and the Fukushima Daiichi Nuclear Power Plant, and river catchments have been identified with labels (Figure 1).

Figure 1: Spatial distribution of the 137Cs inventory in the study catchment (a) and sampling stations of Matsukawa-ura lagoon (b). The spatial distribution of the 137Cs inventory is based on the fourth airborne survey by MEXT (2011). The Voronoi cells were created based on the coordination of Japan Meteorological Agency weather stations (a) and sampling stations (b).

**1. Methods vs. Results**

- Some parameters discussed in Results and Discussion could be introduced in Methods:
  - Kd calculation: Please specify explicitly how Kd was obtained (e.g., Bq m-3 divided by Bq kg-1).
  - o Voronoi partitioning: Mention and briefly describe this procedure in

Methods (currently introduced only in line 199).

• In general, all analytical and data-processing steps used to represent or interpret results should be described in Methods before they appear in the Results section.

The Kd calculation, Voronoi tessellation and mass balance calculation method mentioned in the Results and Discussion have been added in materials and methods (L116-L158). The water catchments basic information tables have been moved to the Supplementary (Table S1).

**Representation of Results**

- **Figure 2:** The statistical plot combines all river catchment data, but much of the underlying data are not shown elsewhere (only partially in Table S1). Consider a clearer, more illustrative figure:
  - o For example, a four-panel plot—one per river—showing monthly dissolved and particulate 137Cs concentrations (Bq m-3) as column plots.
  - This would highlight the key findings: (a) Ume and Nikkeshi contribute most 137Cs to the estuary; (b) fluxes peak in summer.
  - Remaining parameters (Bq kg-1, g m-3, Kd) could either share a secondary axis or be kept in a table placed in the main text rather than in the Supplementary Information.

**Tables:**

Table 1 seems to contain background or supporting data rather than primary results. Consider moving it to the Supplementary Information and bringing Table S1 into the main text.

Since no seasonal trends were observed in the 137Cs concentrations in river water, the currently used box plots were retained (Figure 2), but the time series of dissolved and suspended 137Cs and other parameters were added to the supplementary (Figure S1, Table S2).

**1. Organization of Results and Discussion**

- Section 3.2 (Lagoon results):
  - Subsection 3.2.1 is titled "Relationship between dissolved 137Cs and salinity" but includes all parameters and does not first present the spatial distribution of 137Cs itself. Begin with a clear presentation of particulate and dissolved 137Cs distributions—perhaps using isosurface plots or panel figures similar to those for rivers (even a subset of months for clarity).
  - The investigation of relationships with salinity fits better in the Discussion rather than Results. Splitting Results and Discussion would improve flow and clarity.
- **Figure 3 (salinity relationships):** The panels suggest little to no correlation. Consider exploring:
  - Subsets of data (individual rivers, specific lagoon stations near river mouths, or specific seasons).
  - o If no relationship exists under any subset, move these plots to the Supplementary Information or remove them.
- Section 3.2.2 and Figure 4: Similar concerns apply here—repeated salinity analysis without a clear signal. A deeper targeted analysis or removal is advised.
- **Figure 5:** This clearly shows seasonal variability (higher in summer, lower in winter) and could be placed under a dedicated subsection focused on temporal variability rather than under temperature—salinity relationships.

The title of the subsection 3.2.1 has been changed to "Changes in parameters due to salinity" (L192). The figures of salinity and each parameter have been left intact to clarify whether there is a relationship with salinity (Figure 3), but the figures showing the relationship with salinity at each location have been removed. Instead, figures showing the time series of dissolved and suspended 137Cs has been added (Figure 4), which will lead to the next subsection, "Seasonal variation of dissolved 137Cs in the lagoon". Other parameters have been added in the supplementary (Table S3, S4). In addition, the context has been restructured to make the results and discussion clearer.

A subsection focusing on seasonal variations has been added as subsection 3.2.2, and the effects of water temperature have been integrated into this subsection.

Figure 4: Time series of dissolved 137Cs and particulate 137Cs concentration in Matsukawa-ura lagoon. 137Cs concentrations were decay-corrected to the sampling date. Relative positions of each sampling station are shown in Figure 1b.

**Section 3.3 (Modeling)**

- Add a schematic diagram summarizing the model setup and assumptions for desorption/dissolution of Cs. Values adopte from the literature could be noted on the schematic for transparency.
- Include final model results within the main text (e.g., from Figure S1). Panels a–c of Figure S1 are particularly informative, illustrating that most 137Cs is discharged in August and that the particulate fraction dominates.

We added column plots as Figure 6, showing estimated fluxes from the river. In addition, we added column plots as Figure 7, showing the model results estimating the flux of dissolved 137Cs supplied to the lagoon. These makes the model results easier to understand visually. The original table showing the model results has been reorganized and moved to the supplementary (Table S5).

Figure 6: Fluxes of riverine particulate 137Cs (a), dissolved 137Cs (b) and total 137Cs (c) in each river. Proportion of particulate 137Cs flux in the total 137Cs flux (d). Black bars represent the measurement error of 137Cs.

Figure 7: Potential fluxes of dissolves 137Cs supplied to Matsukawa-ura lagoon and fluxes of dissolved 137Cs from the lagoon to the Pacific Ocean (a). Fluxes of dissolved 137Cs supplied to the lagoon from the rivers, desorbed from riverine particles and bottom sediments in the lagoon (b). Black bars represent the measurement error of 137Cs.

sediments

**1. Broader Perspective and Implications**

- Quantify, if possible, the flux of 137Cs from the lagoon to the Pacific Ocean, addressing one of the paper's stated aims.
- Place findings in a broader context:
  - How significant are lagoon discharges compared with other 137Cs sources to the Pacific?
  - o Are there similar studies (in Japan or elsewhere) that could provide

**comparison?**

- What are the implications of these results for understanding radionuclide transport or for monitoring strategies?
- O Suggest potential next steps or research directions based on the conclusions.

We quantified the flux of 137Cs from the lagoon to the Pacific Ocean (Figure 7a) and compared it with studies reporting fluxes from other rivers in Fukushima Prefecture (Niida et al., 2022) (306-313). Niida et al. (2022) calculated the fluxes of 137Cs flowing out the Pacific Ocean from rivers closer to the Fukushima Daiichi Nuclear Power Plant than Matsukawa-ura lagoon during high flow conditions and found that some of this flux was equivalent to the flux of 137Cs flowing out the Pacific Ocean from Matsukawa-ura lagoon. This highlights the significant impact of the flux of 137Cs flowing out the Pacific Ocean from Matsukawa-ura lagoon compared to other rivers.

Furthermore, because the impact of 137Cs dissolution from bottom sediments is significant (Figure 7b), the report also proposed future monitoring strategies in the Fukushima coastal areas, such as strengthening analysis of marine sediments and their pore water (L329-336).

Table S1
Catchment mean rainfall and flux of river water discharge.

|                                                                                                  |                     | Weather st | ation | Catchment |       |        |          |       |
|--------------------------------------------------------------------------------------------------|---------------------|------------|-------|-----------|-------|--------|----------|-------|
|                                                                                                  |                     | Hippo      | Soma  | Koizumi   | Uda   | Ume    | Nikkeshi | Total |
| Catchment area (km²)                                                                             |                     |            |       | 17.8      | 100.6 | 10.7   | 22.6     | 151.6 |
|                                                                                                  | Forest              |            |       | 51        | 85    | 27     | 62       |       |
| Land use (%)                                                                                     | Paddy field         |            |       | 18        | 5     | 41     | 22       |       |
| Land use (76)                                                                                    | Farmland            |            |       | 6         | 6     | 6      | 4        |       |
|                                                                                                  | Urban area          |            |       | 22        | 1     | 20     | 3        |       |
|                                                                                                  | Koizumi river (KZ)  | 0          | 100   |           |       |        |          |       |
| Ratio to entire catchment (%)                                                                    | Uda river (UD)      | 60         | 40    |           |       |        |          |       |
| Ratio to entire catchinent (70)                                                                  | Ume river (UM)      | 0          | 100   |           |       |        |          |       |
|                                                                                                  | Nikkeshi river (NK) | 0          | 100   |           |       |        |          |       |
|                                                                                                  | Jun-21              | 91         | 74    | 74        | 84    | 74     | 74       |       |
|                                                                                                  | Jul-21              | 165        | 197   | 197       | 177   | 197    | 197      |       |
|                                                                                                  | Aug-21              | 346        | 321   | 321       | 336   | 321    | 321      |       |
|                                                                                                  | Sep-21              | 211        | 195   | 195       | 204   | 195    | 195      |       |
| D : 6 H : 1 - 20 1                                                                               | Oct-21              | 181        | 177   | 177       | 179   | 177    | 177      |       |
| Rainfall in the 30 days prior to sample collection (mm)                                          | Dec-21              | 117        | 64    | 64        | 96    | 64     | 64       |       |
| prior to sample concetion (min)                                                                  | Feb-22              | 21         | 12    | 12        | 17    | 12     | 12       |       |
|                                                                                                  | Jun-22              | 217        | 235   | 235       | 224   | 235    | 235      |       |
|                                                                                                  | Aug-22              | 141        | 99    | 99        | 124   | 99     | 99       |       |
|                                                                                                  | Oct-22              | 54         | 97    | 97        | 71    | 97     | 97       |       |
|                                                                                                  | Feb-23              | 47         | 28    | 28        | 39    | 28     | 28       |       |
|                                                                                                  | Jun-21              |            |       | 0.011     | 0.073 | 0.0068 | 0.014    | 0.11  |
|                                                                                                  | Jul-21              |            |       | 0.030     | 0.15  | 0.018  | 0.039    | 0.24  |
|                                                                                                  | Aug-21              |            |       | 0.050     | 0.29  | 0.030  | 0.063    | 0.44  |
|                                                                                                  | Sep-21              |            |       | 0.030     | 0.18  | 0.018  | 0.038    | 0.26  |
| El                                                                                               | Oct-21              |            |       | 0.027     | 0.16  | 0.016  | 0.035    | 0.23  |
| Flux of river water discharge (×10 6 m 3 12 h -1 ) a | Dec-21              |            |       | 0.010     | 0.083 | 0.0059 | 0.013    | 0.11  |
| (^10 III 12 II )                                                                                 | Feb-22              |            |       | 0.0019    | 0.015 | 0.0011 | 0.0024   | 0.020 |
|                                                                                                  | Jun-22              |            |       | 0.036     | 0.20  | 0.022  | 0.046    | 0.30  |
|                                                                                                  | Aug-22              |            |       | 0.015     | 0.11  | 0.0092 | 0.019    | 0.15  |
|                                                                                                  | Oct-22              |            |       | 0.015     | 0.062 | 0.0090 | 0.019    | 0.10  |
|                                                                                                  | Feb-23              |            |       | 0.0043    | 0.034 | 0.0026 | 0.0055   | 0.047 |

<sup>a Based on the data from Kamo et al.(2014).

Figure S1: Time series of dissolve 137Cs and particulate 137Cs concentration in Matsukawa-ura lagoon and station URS at offshore of the lagoon. 137Cs concentrations were decay-corrected to the sampling date

Table S2  $Analytical\ results\ of\ ^{137}Cs\ concentration\ at\ river\ inflowing\ to\ Matsukawa-ura\ lagoon.$   $^{137}Cs\ activity\ concentrations\ were\ decay-corrected\ to\ the\ sampling\ date.$

| KZ | 2021/6/17               |          |         |            | particles
(Bq kg -1 ) | 137 Cs
(Bq m -3 ) | 137 Cs
(Bq m -3 ) | particle
concentration
(Bq m -3 ) | distribution
coefficient
(L kg -1 ) |
|----|-------------------------|----------|---------|------------|-------------------------------------|--------------------------------------------|--------------------------------------------|----------------------------------------------------|------------------------------------------------------|
|    |                         | 140.9272 | 37.8076 | 0.1        | $1128 ~\pm~ 87$                     | $4.5 ~\pm~ 0.3$                            | $3.5 \pm 0.2$                              | 4.0                                                | 321075                                               |
|    | 2021/7/20               |          |         | 0.2        | $3677\ \pm\ 221$                    | $6.0 \ \pm \ 0.4$                          | $2.7\ \pm\ 0.3$                            | 1.6                                                | 1343027                                              |
|    | 2021/8/27               |          |         | 0.1        | $4835\ \pm\ 854$                    | $1.8~\pm~0.3$                              | $3.7 \ \pm \ 0.3$                          | 0.4                                                | 1290839                                              |
|    | 2021/9/24               |          |         | 0.1        | $1419\ \pm\ 68$                     | $6.5 ~\pm~ 0.3$                            | $2.2 \ \pm \ 0.3$                          | 4.6                                                | 641397                                               |
|    | 2021/10/28              |          |         | 0.1        | $2368\ \pm\ 196$                    | $3.6 ~\pm~ 0.3$                            | $1.2 \ \pm \ 0.3$                          | 1.5                                                | 1976150                                              |
|    | 2021/12/24              |          |         | 0.4        | $1833 \ \pm \ 115$                  | $4.8 \ \pm \ 0.3$                          | $2.6~\pm~0.3$                              | 2.6                                                | 712516                                               |
|    | 2022/2/8                |          |         | 1.0        | $191 \ \pm \ 34$                    | $0.5 ~\pm~ 0.1$                            | $2.7 \ \pm \ 0.3$                          | 2.4                                                | 71815                                                |
|    | 2022/6/23               |          |         | 0.1        | $386\ \pm\ 65$                      | $1.4 \ \pm \ 0.2$                          | $1.0 ~\pm~ 0.3$                            | 3.7                                                | 391027                                               |
|    | 2022/8/30               |          |         | 0.1        | $955\ \pm\ 19$                      | $47.0 \ \pm \ 0.9$                         | $1.8 ~\pm~ 0.3$                            | 49.2                                               | 531742                                               |
|    | 2022/10/27              |          |         | 0.1        | $776 \pm 144$                       | $1.1 ~\pm~ 0.2$                            | $1.2 \ \pm \ 0.3$                          | 1.5                                                | 669661                                               |
|    | 2023/2/20               |          |         | 0.1        | $653 \pm 119$                       | $1.7 ~\pm~ 0.3$                            | $2.1 ~\pm~ 0.3$                            | 2.5                                                | 314126                                               |
| UD | 2021/6/17               | 140.9126 | 37.7902 | 0.1        | $1110 \pm 163$                      | $1.9 \pm 0.3$                              | $2.0 ~\pm~ 0.3$                            | 1.8                                                | 551470                                               |
|    | 2021/7/20               |          |         | 0.1        | $1790 \pm 447$                      | $1.1 \pm 0.3$                              | $2.3 \pm 0.2$                              | 0.6                                                | 777173                                               |
|    | 2021/8/27               |          |         | 0.1        | $2368 \pm 272$                      | $2.4 \pm 0.3$                              | $2.5 \pm 0.3$                              | 1.0                                                | 965710                                               |
|    | 2021/9/24               |          |         | 0.1        | $988 \pm 149$                       | $2.0 \pm 0.3$                              | $1.8 \pm 0.2$                              | 2.0                                                | 550703                                               |
|    | 2021/10/28              |          |         | 0.1        | $1435 \pm 91$                       | $5.1 \pm 0.3$                              | $1.3 \pm 0.3$                              | 3.6                                                | 1110440                                              |
|    | 2021/12/24              |          |         | 0.1        | $4452 \pm 352$                      | $3.1 \pm 0.2$                              | $1.8 \pm 0.3$                              | 0.7                                                | 2419522                                              |
|    | 2022/2/8                |          |         | 0.1        | $3750 \pm 327$                      | $2.9 \pm 0.3$                              | $1.0 \pm 0.2$                              | 0.8                                                | 3698272                                              |
|    | 2022/6/23               |          |         | 0.1        | $376 \pm 122$                       | $0.4 \pm 0.1$                              | $2.2 \pm 0.3$                              | 1.1                                                | 167708                                               |
|    | 2022/8/30               |          |         | 0.1        | $670 \pm 15$                        | $45.7 \pm 1.0$                             | $1.0 ~\pm~ 0.2$                            | 68.2                                               | 648187                                               |
|    | 2022/10/27              |          |         | 0.1        | $1308 \pm 211$                      | $0.7 \pm 0.1$                              | $1.3 \pm 0.3$                              | 0.5                                                | 1016381                                              |
|    | 2023/2/20               |          |         | 0.1        | $2331 \pm 541$                      | $1.2 \pm 0.3$                              | $1.3 \pm 0.3$                              | 0.5                                                | 1813976                                              |
| UM | 2021/6/17               | 140.9494 | 37.7863 | 0.1        | $1496 \pm 30$                       | $45.6 \pm 0.9$                             | $3.4 \pm 0.2$                              | 30.5                                               | 434323                                               |
|    | 2021/7/20               |          |         | 0.1        | $2331 \pm 51$                       | $37.6 \pm 0.8$                             | $2.3 \pm 0.2$                              | 16.1                                               | 992628                                               |
|    | 2021/8/27               |          |         | 0.1        | 9140 ± 384                          | $13.7 \pm 0.6$                             | $3.0 \pm 0.3$                              | 1.5                                                | 3024999                                              |
|    | 2021/9/24               |          |         | 0.1        | 974 ± 29                            | $15.8 \pm 0.5$                             | $3.4 \pm 0.4$                              | 16.2                                               | 285496                                               |
|    | 2021/10/28              |          |         | 0.1        | $1034 \pm 28$                       | $15.8 \pm 0.4$                             | $2.3 \pm 0.3$                              | 15.3                                               | 444498                                               |
|    | 2021/12/24              |          |         | 0.1        | 1244 ± 53                           | $8.7 \pm 0.4$                              | $3.3 \pm 0.3$                              | 7.0                                                | 379072                                               |
|    | 2022/2/8                |          |         | 0.1        | 789 ± 36                            | $7.3 \pm 0.3$                              | $3.0 \pm 0.3$                              | 9.2                                                | 264015                                               |
|    | 2022/6/23               |          |         | 0.1        | $1346 \pm 24$                       | $33.2 \pm 0.6$                             | $2.4 \pm 0.3$                              | 24.7                                               | 554292                                               |
|    | 2022/8/30               |          |         | 0.1        | $2707 \pm 49$                       | 54.1 ± 1.0                                 | $1.7 \pm 0.3$                              | 20.0                                               | 1630890                                              |
|    | 2022/10/27              |          |         | 0.1        | $1014 \pm 60$                       | $5.6 \pm 0.3$                              | $2.6 \pm 0.3$                              | 5.5                                                | 394389                                               |
| NK | 2023/2/20               | 140.9428 | 37.7594 | 0.2        | 454 ± 28                            | $5.5 \pm 0.3$
$10.0 \pm 0.5$            | $1.6 \pm 0.3$
$3.9 \pm 0.2$             | 12.1                                               | 287519                                               |
| NK | 2021/6/17
2021/7/20  | 140.9428 | 37./394 | 0.1
0.1 | $1360 \pm 65$
$560 \pm 88$       |                                            | $3.9 \pm 0.2$
$3.3 \pm 0.2$             | 7.3
2.8                                         | 349932
171579                                     |
|    |                         |          |         | 0.1        | 360 ± 88
1861 ± 210              | $1.5 \pm 0.2$
$3.3 \pm 0.4$             | $3.8 \pm 0.2$ $3.8 \pm 0.3$                |                                                    |                                                      |
|    | 2021/8/27               |          |         |            |                                     |                                            |                                            | 1.8                                                | 490207
690292                                     |
|    | 2021/9/24
2021/10/28 |          |         | 0.1
0.1 | $2176 \pm 74$
$2224 \pm 80$      | $15.3 \pm 0.5$
$12.0 \pm 0.4$           |                                            | 7.1
5.4                                         |                                                      |
|    | 2021/10/28              |          |         | 0.1        | $373 \pm 51$                        | $12.0 \pm 0.4$
$2.1 \pm 0.3$            | $3.2 \pm 0.3$
$3.9 \pm 0.4$             | 5.6                                                | 701280
95406                                      |
|    | 2021/12/24              |          |         | 0.1        | $360 \pm 65$                        | $2.1 \pm 0.3$
$1.5 \pm 0.3$             | $3.9 \pm 0.4$
$3.5 \pm 0.3$             | 4.1                                                | 102398                                               |
|    | 2022/6/23               |          |         | 0.1        | 1208 ± 146                          | $1.3 \pm 0.3$
$2.3 \pm 0.3$             |                                            | 1.9                                                | 513555                                               |
|    | 2022/8/30               |          |         | 0.1        | $1208 \pm 140$ $1702 \pm 32$        | $2.3 \pm 0.3$
$55.8 \pm 1.0$            |                                            | 32.8                                               | 425506                                               |
|    | 2022/10/27              |          |         | 0.1        | $1702 \pm 32$
$1039 \pm 75$      | $3.7 \pm 0.3$                              | $4.0 \pm 0.4$
$4.0 \pm 0.4$             | 3.6                                                | 258130                                               |
|    | 2023/2/20               |          |         | 0.1        | $595 \pm 31$                        | $6.0 \pm 0.3$                              | $3.5 \pm 0.3$                              | 10.1                                               | 168108                                               |

Table S3

Analytical results of 137Cs activity concentration in surface water collected at station UME, OSS, IWS and UDE in Matsukawa-ura lagoon. 137Cs activity concentrations were decay-corrected to the sampling date.

| Station | Sampling date | Long E     | Lat N   | Salinity | 137Cs in suspended
particles
(Bq kg -1 ) | Particulate
137 Cs
(Bq m -3 ) | Dissolved 137 Cs (Bq m -3 ) | Suspended
particle
concentration
(Bq m -3 ) | Apparent
distribution
coefficient
(L kg -1 ) |
|---------|---------------|------------|---------|----------|-----------------------------------------------------------|-----------------------------------------------------------|---------------------------------------------------|-----------------------------------------------------------------|------------------------------------------------------------------|
| UME     | 2021/6/17     | 140.9698   | 37.7869 | 6.0      | 730 ± 14                                                  | $103.3 \pm 2.0$                                           | $23.5 ~\pm~ 0.4$                                  | 141.5                                                           | 31057                                                            |
|         | 2021/7/20     |            |         | 26.8     | $489\ \pm\ 16$                                            | $18.0 ~\pm~ 0.6$                                          | $31.3\ \pm\ 0.7$                                  | 36.8                                                            | 15602                                                            |
|         | 2021/8/27     |            |         | 28.3     | $16433 ~\pm~ 770$                                         | $12.3 \pm 0.6$                                            | $18.0 ~\pm~ 0.5$                                  | 0.7                                                             | 914863                                                           |
|         | 2021/9/24     |            |         | 28.7     | $594 \pm 7$                                               | $123.3 \pm 1.4$                                           | $24.6 \ \pm \ 0.8$                                | 207.6                                                           | 24181                                                            |
|         | 2021/10/28    |            |         | 26.7     | $853 \ \pm \ 26$                                          | $13.2 \pm 0.4$                                            | $14.4 \ \pm \ 0.6$                                | 15.5                                                            | 59062                                                            |
|         | 2021/12/24    |            |         | 23.3     | $2005\ \pm\ 108$                                          | $5.6 \pm 0.3$                                             | $7.2 \pm 0.5$                                     | 2.8                                                             | 278607                                                           |
|         | 2022/2/8      |            |         | 31.0     | $779 ~\pm~ 34$                                            | $7.2 \pm 0.3$                                             | $9.0 ~\pm~ 0.4$                                   | 9.2                                                             | 86841                                                            |
|         | 2022/6/23     |            |         | 28.4     | $1001 \ \pm \ 34$                                         | $12.7 \pm 0.4$                                            | $25.1 \pm 0.8$                                    | 12.7                                                            | 39801                                                            |
|         | 2022/8/30     |            |         | 15.6     | $1369 \ \pm \ 19$                                         | $121.9 ~\pm~ 1.7$                                         | $14.4 \pm 0.6$                                    | 89.0                                                            | 94935                                                            |
|         | 2022/10/27    |            |         | 32.0     | $1509 \ \pm \ 30$                                         | $28.2 \pm 0.6$                                            | $17.7 \pm 0.6$                                    | 18.7                                                            | 85289                                                            |
|         | 2023/2/20     |            |         | 32.5     | $570 \pm 31$                                              | $7.3 \pm 0.4$                                             | $8.3 \pm 0.5$                                     | 12.8                                                            | 68412                                                            |
| OSS     | 2021/6/17     | 140.9792   | 37.7967 | 30.9     | $751 \pm 26$                                              | $13.6 \pm 0.5$                                            | $21.0 \pm 0.4$                                    | 18.2                                                            | 35753                                                            |
|         | 2021/7/20     |            |         | 28.8     | $1708 ~\pm~ 85$                                           | $7.6 \pm 0.4$                                             | $24.0 \pm 0.6$                                    | 4.4                                                             | 71195                                                            |
|         | 2021/8/27     |            |         | 26.1     | $1482 \ \pm \ 141$                                        | $2.8 \pm 0.3$                                             | $12.5 \pm 0.4$                                    | 1.9                                                             | 118460                                                           |
|         | 2021/9/24     |            |         | 28.7     | $1162 \pm 35$                                             | $16.4 \pm 0.5$                                            | $14.0 \pm 0.6$                                    | 14.1                                                            | 83068                                                            |
|         | 2021/10/28    |            |         | 26.8     | $1224 \pm 147$                                            | $2.2 \pm 0.3$                                             | $8.9 \pm 0.5$                                     | 1.8                                                             | 137625                                                           |
|         | 2021/12/24    |            |         | 29.7     | $447 \pm 105$                                             | $0.9 \pm 0.2$                                             | $5.2 \pm 0.4$                                     | 1.9                                                             | 85873                                                            |
|         | 2022/2/8      |            |         | 31.7     | $232 \pm 36$                                              | $1.4 \pm 0.2$                                             | $6.4 \pm 0.4$                                     | 6.1                                                             | 36525                                                            |
|         | 2022/6/23     |            |         | 30.1     | $869 ~\pm~ 68$                                            | $4.0 \pm 0.3$                                             | $17.2 \pm 0.6$                                    | 4.6                                                             | 50544                                                            |
|         | 2022/8/30     |            |         | 9.7      | $503 \pm 21$                                              | $15.8 \pm 0.7$                                            | $9.7 ~\pm~ 0.5$                                   | 31.4                                                            | 51996                                                            |
|         | 2022/10/27    |            |         | 32.6     | $632 \ \pm \ 68$                                          | $2.5 \pm 0.3$                                             | $10.0 \pm 0.5$                                    | 3.9                                                             | 62913                                                            |
|         | 2023/2/22     |            |         | 33.4     | $888 \pm 176$                                             | $1.2 \pm 0.2$                                             | $5.9 \pm 0.4$                                     | 1.3                                                             | 149918                                                           |
| IWS     | 2021/6/17     | 140.9745   | 37.8048 | 31.1     | $1985 \pm 64$                                             | $15.6 \pm 0.5$                                            | $14.8 \pm 8.9$                                    | 7.9                                                             | 134223                                                           |
|         | 2021/7/20     |            |         | 29.0     | $898 \pm 58$                                              | $4.9 \pm 0.3$                                             | $22.6 \pm 0.7$                                    | 5.5                                                             | 39780                                                            |
|         | 2021/8/27     |            |         | 26.7     | $1780 \pm 200$                                            | $2.2 \pm 0.3$                                             | $12.9 \pm 0.5$                                    | 1.3                                                             | 138376                                                           |
|         | 2021/9/24     |            |         | 29.4     | $641 \pm 42$                                              | $4.4 \pm 0.3$                                             | $10.5 \pm 0.5$                                    | 6.9                                                             | 60845                                                            |
|         | 2021/10/28    |            |         | 27.8     | $788 \pm 92$                                              | $2.2 \pm 0.3$                                             | $7.4 \pm 0.5$                                     | 2.8                                                             | 106667                                                           |
|         | 2021/12/24    |            |         | 30.0     | 399 ± 84                                                  | $1.0 \pm 0.2$                                             | $5.4 \pm 0.4$                                     | 2.5                                                             | 74509                                                            |
|         | 2022/2/8      |            |         | 32.0     | 436 ± 143                                                 | $0.5 \pm 0.2$                                             | $3.3 \pm 0.4$                                     | 1.3                                                             | 133630                                                           |
|         | 2022/6/23     |            |         | 28.3     | 398 ± 29                                                  | $4.1 \pm 0.3$                                             | $14.1 \pm 0.5$                                    | 10.4                                                            | 28239                                                            |
|         | 2022/8/30     |            |         | 27.7     | $340 \pm 40$                                              | $2.1 \pm 0.3$                                             | $11.2 \pm 0.5$                                    | 6.3                                                             | 30257                                                            |
|         | 2022/10/27    |            |         | 33.9     | 286 ± 45                                                  | $1.2 \pm 0.2$                                             | $10.1 \pm 0.5$                                    | 4.3                                                             | 28324                                                            |
|         | 2023/2/22     | 1.10.07.12 | 25.0122 | 33.8     | $177 \pm 43$                                              | $0.4 \pm 0.1$                                             | $4.3 \pm 0.3$                                     | 2.3                                                             | 40616                                                            |
| UDE     | 2021/6/17     | 140.9742   | 37.8133 | 32.6     | 751 ± 34                                                  | $8.7 \pm 0.4$                                             | $9.5 \pm 0.4$                                     | 11.6                                                            | 79448                                                            |
|         | 2021/7/20     |            |         | 30.7     | 440 ± 28                                                  | $5.0 \pm 0.3$                                             | $9.4 \pm 0.2$                                     | 11.3                                                            | 46919                                                            |
|         | 2021/8/27     |            |         | 28.9     | 9719 ± 885                                                | $3.2 \pm 0.3$                                             | $10.0 \pm 0.3$                                    | 0.3                                                             | 976626                                                           |
|         | 2021/9/24     |            |         | 10.4     | 751 ± 76                                                  | $2.4 \pm 0.2$                                             | $4.4 \pm 0.4$                                     | 3.2                                                             | 169362                                                           |
|         | 2021/10/28    |            |         | 15.7     | 879 ± 77                                                  | $2.5 \pm 0.2$                                             | $3.8 \pm 0.4$                                     | 2.9                                                             | 234156                                                           |
|         | 2021/12/24    |            |         | 23.7     | 648 ± 114                                                 | $1.2 \pm 0.2$                                             | $3.5 \pm 0.4$                                     | 1.8                                                             | 183115                                                           |
|         | 2022/2/8      |            |         | 27.2     | $307 \pm 83$                                              | $0.5 \pm 0.1$                                             | $2.5 \pm 0.3$                                     | 1.5                                                             | 123684                                                           |
|         | 2022/6/23     |            |         | 31.9     | 595 ± 61                                                  | $2.1 \pm 0.2$                                             | $6.9 \pm 0.4$                                     | 3.5                                                             | 86026                                                            |
|         | 2022/8/30     |            |         | 19.5     | $460 \pm 20$                                              | $7.0 \pm 0.3$                                             | $8.4 \pm 0.4$                                     | 15.2                                                            | 54988                                                            |
|         | 2022/10/27    |            |         | 34.6     | 471 ± 37                                                  | $3.6 \pm 0.3$                                             | $7.3 \pm 0.4$                                     | 7.7                                                             | 64978                                                            |
|         | 2023/2/22     |            |         | 27.2     | $631 \pm 106$                                             | $1.3 \pm 0.2$                                             | $4.4 \pm 0.3$                                     | 2.1                                                             | 143594                                                           |

Table S4

Analytical results of 137Cs activity concentration at Station KZE, WKS, WKO and SKS in Matsukawa-ura lagoon and station URS at offshore of the lagoon. 137Cs activity concentrations were decay-corrected to the sampling date.

| Station | Sampling date          | Long E   | Lat N   | Salinity     | 137Cs in
suspended
particles
(Bq kg -1 ) | Particulate 137 Cs (Bq m -3 ) | Dissolved 137 Cs (Bq m -3 ) | Suspended
particle
concentration
(Bq m -3 ) | Apparent
distribution
coefficient
(L kg -1 ) |
|---------|------------------------|----------|---------|--------------|--------------------------------------------------------------|-----------------------------------------------------|---------------------------------------------------|-----------------------------------------------------------------|------------------------------------------------------------------|
| KZE     | 2021/6/17              | 140.9617 | 37.8208 | 32.3         | $644 \pm 32$                                                 | $6.8 \pm 0.3$                                       | 20.3 ± 0.4                                        | 10.6                                                            | 31758                                                            |
|         | 2021/7/20              |          |         | 30.4         | $361 \ \pm \ 34$                                             | $2.9 \pm 0.3$                                       | $11.5 ~\pm~ 0.5$                                  | 8.0                                                             | 31282                                                            |
|         | 2021/8/27              |          |         | 30.4         | $2240\ \pm\ 171$                                             | $4.3 ~\pm~ 0.3$                                     | $12.3 ~\pm~ 0.3$                                  | 1.9                                                             | 181685                                                           |
|         | 2021/9/24              |          |         | 29.6         | $466\ \pm\ 14$                                               | $14.5~\pm~0.4$                                      | $11.8~\pm~0.5$                                    | 31.0                                                            | 39624                                                            |
|         | 2021/10/28             |          |         | 30.7         | $704\ \pm\ 70$                                               | $2.5~\pm~0.2$                                       | $6.8 ~\pm~ 0.5$                                   | 3.5                                                             | 102786                                                           |
|         | 2021/12/24             |          |         | 30.0         | $193~\pm~63$                                                 | $0.4\ \pm\ 0.1$                                     | $4.7 \ \pm \ 0.3$                                 | 2.2                                                             | 41199                                                            |
|         | 2022/2/8               |          |         | 32.3         | $555\ \pm\ 153$                                              | $0.8 \ \pm \ 0.2$                                   | $4.7 \ \pm \ 0.3$                                 | 1.5                                                             | 118216                                                           |
|         | 2022/6/23              |          |         | 27.8         | $905 ~\pm~ 42$                                               | $7.5~\pm~0.3$                                       | $15.5~\pm~0.6$                                    | 8.3                                                             | 58171                                                            |
|         | 2022/8/30              |          |         | 29.7         | $483\ \pm\ 25$                                               | $6.5 \ \pm \ 0.3$                                   | $12.0\ \pm\ 0.5$                                  | 13.4                                                            | 40271                                                            |
|         | 2022/10/27             |          |         | 34.1         | $375\ \pm\ 72$                                               | $1.4 ~\pm~ 0.3$                                     | $7.7 ~\pm~ 0.5$                                   | 3.6                                                             | 48482                                                            |
|         | 2023/2/22              |          |         | 33.9         | $401 \ \pm \ 76$                                             | $1.1~\pm~0.2$                                       | $5.0 ~\pm~ 0.4$                                   | 2.8                                                             | 79378                                                            |
| WKS     | 2021/6/17              | 140.9807 | 37.8199 | 29.0         | $160~\pm~14$                                                 | $4.1 \ \pm \ 0.3$                                   | $13.4 \ \pm \ 0.6$                                | 25.5                                                            | 11979                                                            |
|         | 2021/7/20              |          |         | 31.5         | $171\ \pm\ 49$                                               | $1.0 ~\pm~ 0.3$                                     | $7.5 ~\pm~ 0.2$                                   | 6.0                                                             | 22819                                                            |
|         | 2021/8/27              |          |         | 29.2         | $1790 \pm 254$                                               | $2.0 ~\pm~ 0.3$                                     | $13.6 ~\pm~ 0.4$                                  | 1.1                                                             | 131290                                                           |
|         | 2021/9/24              |          |         | 29.6         | $583 \pm 54$                                                 | $3.4 \ \pm \ 0.3$                                   | $10.9 ~\pm~ 0.5$                                  | 5.8                                                             | 53455                                                            |
|         | 2021/10/28             |          |         | 31.4         | $1086 \ \pm \ 106$                                           | $3.3 \ \pm \ 0.3$                                   | $6.2 \ \pm \ 0.5$                                 | 3.0                                                             | 175107                                                           |
|         | 2021/12/24             |          |         | 29.6         | $557 \pm 165$                                                | $0.7 \pm 0.2$                                       | $5.6 \pm 0.4$                                     | 1.3                                                             | 100232                                                           |
|         | 2022/2/8               |          |         | 32.3         | $347 \pm 107$                                                | $0.5 \pm 0.1$                                       | $4.9 \pm 0.3$                                     | 1.3                                                             | 71415                                                            |
|         | 2022/6/23              |          |         | 31.8         | $404 \pm 58$                                                 | $0.8 \pm 0.1$                                       | $8.4 \pm 0.5$                                     | 1.9                                                             | 48078                                                            |
|         | 2022/8/30              |          |         | 26.1         | $407 \pm 65$                                                 | $1.5 \pm 0.2$                                       | $11.1 \pm 0.5$                                    | 3.7                                                             | 36777                                                            |
|         | 2022/10/27             |          |         | 34.2         | $9604 \pm 261$                                               | $15.4 \pm 0.4$                                      | $7.9 \pm 0.5$                                     | 1.6                                                             | 1209155                                                          |
|         | 2023/2/20              |          |         | 34.6         | $204 \pm 29$                                                 | $0.2 \pm 0.0$                                       | $3.5 \pm 0.4$                                     | 1.0                                                             | 57484                                                            |
| WKO     | 2021/6/17              | 140.9756 | 37.8219 | 31.4         | $699 \pm 71$                                                 | $2.3 \pm 0.2$                                       | $7.2 \pm 0.2$                                     | 3.3                                                             | 96859                                                            |
|         | 2021/7/20              |          |         | 31.3         | $287 \pm 20$                                                 | $4.0 \pm 0.3$                                       | $6.5 \pm 0.2$                                     | 13.8                                                            | 44141                                                            |
|         | 2021/8/27              |          |         | 30.0         | 553 ± 77                                                     | $1.8 \pm 0.2$                                       | $8.0 \pm 0.3$                                     | 3.2                                                             | 68885                                                            |
|         | 2021/9/24              |          |         | 29.6         | $1097 \pm 102$                                               | $2.3 \pm 0.2$                                       | $10.4 \pm 0.5$                                    | 2.1                                                             | 105583                                                           |
|         | 2021/10/28             |          |         | 30.9         | $1202 \pm 192$                                               | $1.6 \pm 0.2$                                       | $6.8 \pm 0.5$                                     | 1.3                                                             | 176470                                                           |
|         | 2021/12/24             |          |         | 30.2         | $1007 \pm 218$                                               | 1.1 ± 0.2                                           | $5.1 \pm 0.4$                                     | 1.1                                                             | 199352                                                           |
|         | 2022/2/8               |          |         | 32.9         | 100 ± 29                                                     | $0.1 \pm 0.0$                                       | $2.9 \pm 0.3$                                     | 1.5                                                             | 34702                                                            |
|         | 2022/6/23              |          |         | 32.0         | 590 ± 106                                                    | $1.3 \pm 0.2$                                       | $5.6 \pm 0.4$                                     | 2.2                                                             | 106005                                                           |
|         | 2022/8/30              |          |         | 29.8         | 871 ± 157
1044 ± 81                                       | $1.2 \pm 0.2$
$3.8 \pm 0.3$                      | $7.6 \pm 0.5$                                     | 1.4                                                             | 114686                                                           |
|         | 2022/10/27             |          |         | 33.3         |                                                              |                                                     | $7.1 \pm 0.4$                                     | 3.7                                                             | 146499                                                           |
| SKS     | 2023/2/22              | 140.9774 | 37.8277 | 34.4
31.4 | $765 \pm 220$ $190 \pm 42$                                   | $0.4 \pm 0.1$
$0.9 \pm 0.2$                      | $4.1 \pm 0.4$
$7.6 \pm 0.6$                    | 0.6
4.5                                                      | 185232                                                           |
| SKS     | 2021/6/17
2021/7/20 | 140.9774 | 37.6277 | 29.6         | 190 ± 42
199 ± 35                                         | $0.9 \pm 0.2$
$0.9 \pm 0.2$                      | $14.2 \pm 0.3$                                    | 4.7                                                             | 25102
14047                                                   |
|         | 2021/7/20              |          |         | 30.0         | $199 \pm 35$
$2024 \pm 263$                               | $0.9 \pm 0.2$ $1.7 \pm 0.2$                         | $7.9 \pm 0.4$                                     | 0.9                                                             | 255302                                                           |
|         | 2021/9/24              |          |         | 29.0         | 2024 ± 203
208 ± 44                                       | $0.4 \pm 0.1$                                       | $7.3 \pm 0.4$
$7.1 \pm 0.4$                    | 2.1                                                             | 29456                                                            |
|         | 2021/10/28             |          |         | 28.7         | $631 \pm 160$                                                | $0.9 \pm 0.2$                                       | $7.2 \pm 0.5$                                     | 1.4                                                             | 87242                                                            |
|         | 2021/12/24             |          |         | 29.4         | $387 \pm 98$                                                 | $0.6 \pm 0.2$                                       | $6.8 \pm 0.5$                                     | 1.6                                                             | 57160                                                            |
|         | 2022/2/8               |          |         | 32.3         | $522 \pm 127$                                                | $0.9 \pm 0.2$                                       | $4.0 \pm 0.4$                                     | 1.7                                                             | 130624                                                           |
|         | 2022/6/23              |          |         | 30.1         | $749 \pm 102$                                                | $1.8 \pm 0.2$                                       | 9.4 ± 0.5                                         | 2.5                                                             | 80067                                                            |
|         | 2022/8/30              |          |         | 29.3         | $148 \pm 37$                                                 | $0.2 \pm 0.1$                                       | $6.2 \pm 0.4$                                     | 1.6                                                             | 23764                                                            |
|         | 2022/10/27             |          |         | 34.1         | 551 ± 62                                                     | 2.3 ± 0.3                                           | 9.0 ± 0.4                                         | 4.3                                                             | 61196                                                            |
|         | 2023/2/22              |          |         | 33.7         | $1256 \pm 329$                                               | $0.9 \pm 0.2$                                       | $4.4 ~\pm~ 0.4$                                   | 0.7                                                             | 286949                                                           |
| URS     | 2021/8/2               | 141.0039 | 37.8000 | 32.0         | ND                                                           | ND                                                  | 3.1 ± 0.2                                         | 3.0                                                             | -                                                                |
|         | 2021/10/14             |          |         | 30.5         | $167 \pm 53$                                                 | $0.2 \pm 0.1$                                       | $4.8 ~\pm~ 0.4$                                   | 1.0                                                             | 34840                                                            |
|         | 2022/2/1               |          |         | 34.4         | $403 \pm 76$                                                 | $0.4 \pm 0.1$                                       | $3.1 \pm 0.4$                                     | 1.0                                                             | 131808                                                           |
|         | 2022/6/1               |          |         | 33.7         | $122\ \pm\ 28$                                               | $0.4 \ \pm \ 0.1$                                   | $2.8~\pm~0.3$                                     | 3.2                                                             | 43007                                                            |
|         | 2022/8/3               |          |         | 32.7         | ND                                                           | ND                                                  | $2.7 ~\pm~ 0.3$                                   | 0.8                                                             | -                                                                |
|         | 2022/10/3              |          |         | 33.4         | $247\ \pm\ 61$                                               | $0.8 ~\pm~ 0.2$                                     | $2.5~\pm~0.3$                                     | 3.2                                                             | 98023                                                            |
|         | 2023/2/7               |          |         | 34.3         | $175~\pm~48$                                                 | $0.2 \ \pm \ 0.1$                                   | $2.2 ~\pm~ 0.3$                                   | 1.2                                                             | 77673                                                            |

Table S5

Mass balance of dissolved 137Cs in Matsukawau-ra lagoon.

|        | Time
after the
accident | Input of
seawater into
Matsukawa-
ura lagoon a | Input of river
water into
Matsukawa-
ura lagoon | Dissolved 137 Cs
in coastal water
(station URS) | Weighted
mean
dissolved
137 Cs in
lagoon | Potential flux of 137 Cs | Flux of riverine dissolved 137Cs | Desorable fracton of riverine particulate 137Cs b | Flux of 137 Cs
desorbed
from bottom
sediment | Flux of
dissolved
137 Cs
outflowing
to Pacific
Ocean |
|--------|-------------------------------|--------------------------------------------------------------------|----------------------------------------------------------|------------------------------------------------------------------|-----------------------------------------------------------------|-------------------------------------|----------------------------------|--------------------------------------------------------------|------------------------------------------------------------------|--------------------------------------------------------------------------------|
|        | day                           | $Mm^3 12 h^{-1}$                                                   | $\text{Mm}^3 12  \text{h}^{-1}$                          | Bq m -3                                               | Bq m -3                                              | MBq 12 h -1              | MBq 12 h -1           | MBq 12 h -1                                       | MBq 12 h -1                                           | MBq 12 h -1                                                         |
| Jun-21 | 3751                          | 7.2                                                                | 0.11                                                     | 2.2 - 4.8                                                        | 17.0                                                            | 88 - 106                            | 0.27                             | 0.036                                                        | 87 - 106                                                         | 124                                                                            |
| Jul-21 | 3784                          | 7.2                                                                | 0.24                                                     | 2.2 - 4.8                                                        | 19.0                                                            | 102 - 121                           | 0.61                             | 0.060                                                        | 102 - 120                                                        | 142                                                                            |
| Aug-21 | 3822                          | 7.2                                                                | 0.44                                                     | 2.2 - 4.8                                                        | 13.1                                                            | 60 - 78                             | 1.23                             | 0.077                                                        | 58 - 77                                                          | 100                                                                            |
| Sep-21 | 3850                          | 7.2                                                                | 0.26                                                     | 2.2 - 4.8                                                        | 13.5                                                            | 62 - 81                             | 0.57                             | 0.078                                                        | 62 - 80                                                          | 100                                                                            |
| Oct-21 | 3884                          | 7.2                                                                | 0.23                                                     | 2.2 - 4.8                                                        | 8.5                                                             | 27 - 45                             | 0.38                             | 0.086                                                        | 26 - 45                                                          | 63                                                                             |
| Dec-21 | 3941                          | 7.2                                                                | 0.11                                                     | 2.2 - 4.8                                                        | 5.4                                                             | 4.7 - 23                            | 0.25                             | 0.021                                                        | 4.4 - 23                                                         | 40                                                                             |
| Feb-22 | 3987                          | 7.2                                                                | 0.02                                                     | 2.2 - 4.8                                                        | 5.3                                                             | 3.4 - 22                            | 0.03                             | 0.0031                                                       | 3.3 - 22                                                         | 38                                                                             |
| Jun-22 | 4122                          | 7.2                                                                | 0.30                                                     | 2.2 - 4.8                                                        | 15.0                                                            | 74 - 93                             | 0.64                             | 0.053                                                        | 73 - 92                                                          | 113                                                                            |
| Aug-22 | 4190                          | 7.2                                                                | 0.15                                                     | 2.2 - 4.8                                                        | 11.0                                                            | 44 - 63                             | 0.23                             | 0.40                                                         | 44 - 62                                                          | 81                                                                             |
| Oct-22 | 4248                          | 7.2                                                                | 0.10                                                     | 2.2 - 4.8                                                        | 10.6                                                            | 42 - 61                             | 0.20                             | 0.010                                                        | 42 - 60                                                          | 77                                                                             |
| Feb-23 | 4366                          | 7.2                                                                | 0.05                                                     | 2.2 - 4.8                                                        | 5.5                                                             | 4.8 - 24                            | 0.08                             | 0.0052                                                       | 4.8 - 23                                                         | 40                                                                             |

<sup>a Kamo et al.(2014).

<sup>b Assumed that 5.5% of riverine particulate 137Cs were desorbed as dissolved 137Cs (Takata et al., 2021)

---

## Author Comment (AC2)

**General Comments**

This study focuses on Matsukawa-ura, a brackish lagoon, where the authors conducted observations of dissolved and particulate ^137Cs concentrations in lagoon and inflowing river waters during 2021–2023, approximately ten years after the Fukushima Daiichi Nuclear Power Plant accident. The study demonstrates the seasonal dependence of dissolved ^137Cs concentrations in the lagoon and estimates particulate and dissolved fluxes from rivers. The results suggest that riverine fluxes alone cannot explain the elevated dissolved ^137Cs concentrations in the lagoon, and the authors cite previous studies to point out the possible contribution from bottom sediments. The observational dataset itself is valuable and provides important insights for understanding the long-term dynamics of radiocesium in Matsukawa-ura.

However, the section on mass flux analysis contains fundamental problems, and in its current form, the conclusions cannot be supported. The flux estimation includes many questionable assumptions and methodological flaws. I believe that this part requires major revision or should be removed entirely.

While the observational data are highly valuable, the mass balance analysis is methodologically and conceptually inappropriate and does not support the conclusions. I strongly recommend removing or substantially revising this section. If the manuscript focuses instead on the observational results and their comparison with previous studies, it would represent a useful and significant contribution.

We thank the reviewer for your time and constructive feedback. The comparison with previous studies are going to be added for our study being a useful and contribution for the recovery of the coastal areas in this estuarine area. Please see our responses below.

**Specific Comments**

**Mass balance section**

L244: The authors estimated the supply flux from sediments by using the tidal prism to calculate "the volume of seawater inflow per 12 hours," and assuming that this inflowing water uniformly reaches the lagoon's mean concentration within 12 hours. The difference was then attributed to sedimentary fluxes. However, this estimation has several critical flaws:

(1) The tidal prism does not represent a "pure inflow" of new seawater. In reality, inflow and outflow occur simultaneously, and the exchange efficiency is less than 100%.

As suggested by the reviewer, the seawater that flows into Matsukawa-ura lagoon is not 100% replaced, but I wanted to suggest a qualitative trend that there is a seasonality. We have addressed the lack of this as follows:

"Although the tidal prism does not fully represent the entire exchange of estuarine water with oceanic seawater entering from outside the lagoon, it is used here—based on the simplified assumption that pure seawater flows in from the open ocean—to help understand the monthly variation in the influx of dissolved 137Cs and the key factors maintaining relatively high 137Cs concentrations in the lagoon" in the main text.

- (2) The assumption that inflowing water reaches the lagoon-wide average concentration uniformly within 12 hours is unrealistic. Observations clearly show higher concentrations in the inner lagoon, demonstrating strong spatial heterogeneity.

  The water depth of Matsukawa-ura lagoon is very shallow, averaging about 1m, and it has been indicated that about half of the water is exchanged during one tidal cycle (Kohata et al., 2003). For the simplicity in clarifying the seasonality of dissolved 137Cs in the lagoon, we used the average concentration of dissolved 137Cs among several sampling stations which could might represent the dissolved 137Cs levels in the area of the lagoon. Actually, in February 2022 and 2023, dissolved 137Cs concentrations ranged 2.5-9.0 and 3.5-8.3 Bq m-3 with small variation, respectively, with weighted mean dissolved 137Cs concentrations of 5.3 and 5.5 Bq m-3, but, weighted mean dissolved 137Cs in June 2022 were 17.0 and 15.0 Bq m-3, respectively, about three times higher than winter season. These results suggest that seasonal variations are more influential than variations in concentrations between areas within the lagoon.
- (3) The authors assume a constant tidal prism exchange volume throughout the year, yet still discuss seasonal (summer vs. winter) differences. This is logically inconsistent.

For these reasons, the flux estimation is not valid and should be considered inappropriate.

This study focused on the seasonality of dissolved 137Cs in Matsukawa-ura lagoon, and the calculations showed that dissolved 137Cs in Matsukawa-ura lagoon is seasonal and would like to suggest the influence of water temperature. In this study, we measured the water depth at the station WKO, where connects with Pacific Ocean, from December 29, 2021. The average water depth during the study period was 1.9 m, with a maximum of 2.2 m and a minimum of 1.7 m, and the difference between the maximum and minimum values was about 50 cm (Figure a). Therefore, the exchange volume in Matsukawa-ura lagoon was assumed to be constant throughout the year, making it possible to compare between

seasons.

Figure a. Time series changes in water depth at Station WKO from December 29, 2021.

L260: The method used to estimate the decrease of ^137Cs in bottom sediments is unclear. Given that the mass flux calculation itself is subject to large uncertainties, presenting the results as quantified values—and further highlighting them in the abstract—risks giving readers an impression of unwarranted accuracy. Considering the likelihood of these numbers being cited in future work, the current presentation lacks scientific integrity. This section should be deleted or, at the very least, thoroughly rewritten.

We will remove the section on estimating the decrease of 137Cs in sediments, suggest only seasonality. The abstract will be revised accordingly to exclude quantitative values and instead emphasize the seasonality.

L31 : Sanial et al. (2017) is not a study of sedimentary fluxes and its citation here is inappropriate. The positioning of this reference should be reconsidered.

Introduction: Kambayashi et al. (2021) is a highly relevant prior study and should be cited in the Introduction.

We remove Sanial et al. (2017) as a citation in the introduction and add Kambayashi et al. (2021) as a new citation in the introduction and discussion for the comparison with our study. Kambayashi et al. (2021), studied in Matsukawa-ura lagoon in 2014-2016, calculated the mass balance of dissolved 137Cs in Matsukawa-ura lagoon, and suggested that supply from porewater in bottom sediments increased in the summer, and the fluxes of dissolved 137Cs from bottom sediments were 287-293 MB day-1 in summer. The estimated fluxes of

dissolved 137Cs from bottom sediments during the summer were consistent with our results. These results suggest that the amount of 137Cs deposited in marine sediments in the early stages of the accident and dissolved and dispersed into the seawater may not change significantly, whether three or ten years have passed since the accident. Thus, as more time passes since the accident and the supply of 137Cs to coastal waters from rivers decreases due to the decommissioning of contaminated soils in their catchment areas, the contribution of the dissolution of this nuclide from bottom sediments may have continued.

Fig. 4: The spatial arrangement of sampling sites is difficult to understand. I recommend improving the map by adding scale bars, landmarks, or other visual cues to make the locations more intuitive.

We have done.

**Reference**

Kohata, K.; Hiwatari, T.; Hagiwara, T. Natural water-purification system observed in a shallow coastal lagoon: Matsukawa-ura, Japan. Marine Pollution Bulletin. 2003, 47, 148-154. https://doi.org/10.1016/S0025-326X(03)00055-9

Kambayashi, S.; Zhang, J.; Narita, H. Significance of Fukushima-derived radiocaesium flux via river-estuary-ocean system. Sci. Total Environ. 2021, 793, 148456. https://doi.org/10.1016/j.scitotenv.2021.148456